# Emergence of Hidden Capabilities:
# Exploring Learning Dynamics in Concept Space

**Core Francisco Park**[*1,2,3], **Maya Okawa**[*1,3], **Andrew Lee**[4]

**Hidenori Tanaka**[†1,3], **Ekdeep Singh Lubana**[†1,3]

[1]CBS-NTT Program in Physics of Intelligence, Harvard University
[2]Department of Physics, Harvard University
[3]Physics & Informatics Laboratories, NTT Research, Inc.
[4]EECS Department, University of Michigan, Ann Arbor

## Abstract

Modern generative models demonstrate impressive capabilities, likely stemming from an ability to identify and manipulate abstract concepts underlying their training data. However, fundamental questions remain: what determines the concepts a model learns, the order in which it learns them, and its ability to manipulate those concepts? To address these questions, we propose analyzing a model's learning dynamics via a framework we call the *concept space*, where each axis represents an independent concept underlying the data generating process. By characterizing learning dynamics in this space, we identify how the speed at which a concept is learned, and hence the order of concept learning, is controlled by properties of the data we term *concept signal*. Further, we observe moments of *sudden turns* in the direction of a model's learning dynamics in concept space. Surprisingly, these points precisely correspond to the emergence of *hidden capabilities*, i.e., where latent interventions show the model possesses the capability to manipulate a concept, but these capabilities cannot yet be elicited via naive input prompting. While our results focus on synthetically defined toy datasets, we hypothesize a general claim on *emergence of hidden capabilities* may hold: generative models possess latent capabilities that emerge suddenly and consistently during training, though a model might not exhibit these capabilities under naive input prompting.

## 1 Introduction

Modern generative models, such as text-conditioned diffusion models, show unprecedented capabilities [1–8]. These abilities have led to use of such models in applications as valuable as training control policies for robotics [9–11] and models for weather forecasting [12], to as drastic as campaigning in democratic elections [13, 14]. Similar claims can be made for generative models of other modalities, e.g., large language models (LLMs) [15–18], speech and audio models [14, 19], or even systems designed for enabling scientific applications such as drug discovery [20, 21]. Arguably, acquiring such general capabilities requires for models to internalize the data-generating process and disentangle the concepts (aka latent variables or factors of variation) underlying it [22, 23]. Flexibly manipulating these concepts can then enable generation of novel samples that are entirely out-of-distribution (OOD) with respect to the ones used for training [24–28].

As shown in prior work, modern generative models do exhibit signs of disentangling concepts underlying the data generating process and learning of corresponding capabilities to manipulate said

---

* equal contribution, † equal advising (see detailed list). Email: corefranciscopark@g.harvard.edu, ajyl@umich.edu, {mayaokawa, ekdeeplubana, hidenori_tanaka}@fas.harvard.edu

38th Conference on Neural Information Processing Systems (NeurIPS 2024).

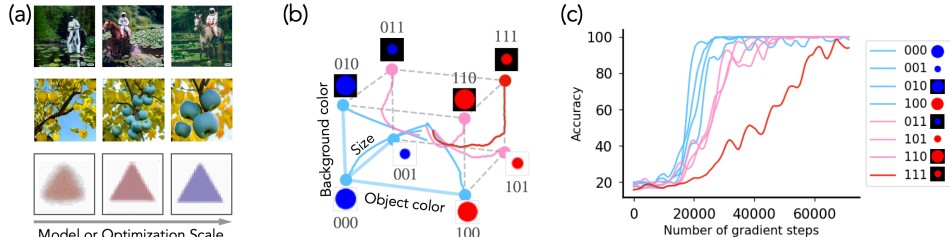

Figure 1: **Concept Learning Geometry underlies emergence.** (a) Top: A multimodal model learns to generate the concepts in the order of "astronaut", "horse", and finally "riding" as it scales up (adapted from Yu et al.[45]). Middle: "blue square apple" is generated in the order of "apple", "blue", and "square"" (adapted from Li et al.[46]). Bottom: Despite its simplicity, our model trained on synthetic data shows *concept learning dynamics* where it first learns "shape" and then "color". (b) Concept space is an abstract coordinate space where individual axes correspond to different concepts and a given point corresponds to a "concept class", i.e., a predefined collection of concepts (e.g., `large blue circles` on bottom left corner). Traversal along axes of the concept space yield change in a specific property of the sample (e.g., going from `large blue circle` to `large red circle` along `object color` axis). Trajectories show a model's dynamics in concept space for learning to generate classes shown in-distribution (blue nodes) versus out of distribution (pink / red nodes). As we show, dynamics in concept space are highly interpretable, enabling precise comments on which concepts the model learns first, why, and what order it follows. (c) Measuring how accurately a model generates samples from a given concept class, showing an order of concept learning: first background color is learned, then size, and then object color.

concepts [29–37]. At the same time, contemporary work has argued that models' capabilities can be unreliable, arbitrarily failing for a given input and succeeding on another [38–44]. Thus, critical questions remain on how generative models acquire their capabilities (see Fig. 1): what determines whether the model will disentangle a concept and learn the capability to manipulate it; are all concepts and corresponding capabilities learned at the same time; and is there a structure to the order of concept learning such that, given insufficient time, the model learns capabilities to manipulate only a subset of concepts but not others?

**This work.** To address the questions above, we propose to analyze a model's *learning dynamics at the granularity of concepts*. Since performing such an analysis on realistic, off-the-shelf datasets can be challenging, we develop synthetic toy datasets of 2D objects with different concepts (e.g., shape, size, color) that give us thorough control over the data-generating process and allow for an exhaustive characterization of the model's learning dynamics (see Fig. 1). Our contributions are as follows.

- **Introducing Concept Space.** We propose to evaluate a model's learning dynamics in the *concept space*—an abstract coordinate system whose axes correspond to specific concepts underlying the data-generating process. We instantiate a notion of underspecification in concept space and establish its effects on a model's (in)ability to disentangle concepts.

- **Concept Signal Dictates Speed of Learning.** We find the speed at which a model disentangles a concept and learns the capability to manipulate it is dictated by the sensitivity of the data-generating process to changes in values of said concept—a quantity we call *concept signal*. We show concept signals shape the geometry of a learning trajectory and hence control the overall order of concept learning.

- **Sudden Transitions in Concept Learning.** We use analytical curves to explain the phenomenology of learning dynamics in concept space, showing the dynamics can be divided into two broad phases: **(P1)** learning of a *hidden capability*, whereby even if the model does not produce the desired output for a given input, there exist systematic latent interventions that lead the model to generate the desired output, and **(P2)** learning to generate the desired output from the input space.

Overall, while our results focus on a toy synthetic task and text-to-image diffusion models, we hypothesize a broader claim on *hidden emergence of latent capabilities* holds true: generative models possess latent capabilities that are learned suddenly and consistently during training, but these capabilities are not immediately apparent since prompting the model via the input space may not elicit them, hence hiding how "competent" the model actually is [47, 48]. Empirically, signs of "hidden capabilities" have already been shown in generative models at scale [45, 49–53], and our results help provide a formal framework to partially ground such results.

## 2 Related Work

**Concept learning.** The term concepts as used in this work is broadly equivalent to the notion of factors of variation from prior work on disentangled representation learning [23, 30, 54–61], and is motivated by use of the term in a similar sense in cognitive science [62–67]. The focus of literature on disentanglement has been to prove identifiability results, e.g., when will a generative model trained on a dataset learn to invert the data-generating process, hence identifying the latent variables, i.e., concepts, that underlie it. Given the success of modern generative models, we argue, despite impossibility results from prior work, these models are in fact learning to disentangle some, if not all, concepts. Precisely what guides which concepts are disentangled however requires studying the learning dynamics—the target of our work.

**Interpretability.** A growing line of papers demonstrate highly interpretable representations of intuitive concepts exist in generative models, especially LLMs [68–74]. Similar work in this vein in image diffusion models has found semantic representations in various components of the model [37, 52, 75–77]: e.g., existence of linear representations for concepts such as 3D depth or object versus background distinctions [78, 79] . These papers further bolster our argument, modern generative models are truly inverting the data-generating process and identifying the concepts underlying it.

**Competence vs. performance.** In cognitive science, a system's competence on a task is often contrasted with its performance [47, 48, 80–83]: competence is the system's *possession of a capability* (e.g., to converse in a language) and performance is system's *use of that capability in concrete situations* [80]. For example, a bilingual person may generally converse in their primary language $\mathcal{L}$, despite possessing knowledge of another language $\mathcal{L}'$, unless it is crucial to use the latter—clearly, they are competent in both languages, but gauging their performance on $\mathcal{L}'$ requires appropriately "prompting" them to use it. One can analogize this distinction with remarks on a neural network possessing a capability versus us being able to elicit it on predefined benchmarks and measure their performance [84–87]. For example, on the BigBench benchmark [88], LLMs were shown to have perform poorly on several tasks, but follow up work [89] showed mere chain-of-thought prompting [49, 50] leads to huge boosts on all tasks. This indicates the evaluated models were in fact "competent", but inappropriate prompting led to undermining how "performant" they are.

## 3 Concept Space: A Framework for Analyzing Concept Learning Dynamics

We first formally introduce our framework of *concept spaces*. See Fig. 12 for a schematic. Motivated by a rich body of literature on disentangled representation learning [23, 30, 54–60], this framework allows us to systematically analyze how different concepts underlying the data generating process and corresponding capabilities to manipulate them are learned during training. We highlight that since our primary empirical focus in this paper will be an abstraction of text-to-image generative models, parts of the framework will be specifically instantiated with text-to-image generation tasks in mind. To this end, we note our model class of interest is a generative model $F$ that is trained using conditioning information $h$ to produce images $y$. For example, $F$ may be instantiated using a diffusion model that uses embeddings $h$ of textual descriptions of a scene to produce images $y$. We next define a *concept space*.

**Definition 1.** *(Concept Space.) Consider an invertible data-generating process $\mathcal{G} : \mathcal{Z} \to \mathcal{X}$ that samples vectors $z \sim P(\mathcal{Z})$ from a vector space $\mathcal{Z} \subset \mathbb{R}^d$ and maps them to the observation space $\mathcal{X} \in \mathbb{R}^n$. We assume the sampling prior is factorizable, i.e., $P(z \in \mathcal{Z}) = \Pi_{i=1}^d P(z_i)$, and individual dimensions of $\mathcal{Z}$ correspond to semantically meaningful concepts. Then, a concept space $\mathcal{S}$ is defined as the multidimensional space composed of all possible concept vectors $z$, i.e., $\mathcal{S} := \{z \mid z \sim P(\mathcal{Z})\}$*

As an example, consider a concept space defined using three concepts, say, $z_1 = \texttt{shape}$, $z_2 = \texttt{size}$, and $z_3 = \texttt{color}$ that maps to a dataset of images with objects of different combinations of shapes, sizes, and colors (see Fig. 2). The assumption that concepts are independently distributed implies one can intervene on a given concept without affecting the other ones. For example, given a sample from the dataset above, the concept $\texttt{color}$ in an image can be altered by changing the value of the relevant latent variable and mapping it to the image space via the data-generating process. Now, using a mixing function $\mathcal{M}$ that yields conditioning information $h := \mathcal{M}(z)$, we can train a conditional generative model and define a notion of capabilities relevant to our work as follows.

**Definition 2.** *(Capability.)* *A concept class $\mathcal{C}$ denotes the set of concept vectors $z_\mathcal{C}$ such that a subset of dimensions of these vectors are fixed to predefined values. Classes $\mathcal{C}$ and $\mathcal{C}'$ are said to differ in the $k^{th}$ concept if $\forall z \in z_\mathcal{C}$, there exists $z' \in z_{\mathcal{C}'}$ with $z[k] \neq z'[k]$ and $z[i] = z'[i]$ for $i \neq k$. We say a model possesses the "capability to alter the $k^{th}$ concept" if for any class $\mathcal{C}$ whose samples were seen during training, the model can produce samples from class $\mathcal{C}'$ that differs from $\mathcal{C}$ in the $k^{th}$ concept.*

Intuitively, the definition above comments on whether the model can flexibly manipulate concepts of classes seen during training to produce samples from classes that were not seen, i.e., classes that are entirely out-of-distribution. As an example, consider a concept space with `shape`, `color`, and `size` as concepts. If `shape` and `color` are fixed to `circle` and `blue` respectively, we get the class of `blue circles`; i.e., $\forall z \in z_{\texttt{blue circles}}$, the first and second dimension respectively take on values that correspond to the shape `circle` and color `blue`. Then, given a conditional diffusion model that was shown `blue circles` during training, we will say this model possesses the capability to alter the concept `color` if it can produce samples from concept classes with the same shape as `circles`, but different colors (e.g., `red` or `green circles`). Analyzing learning dynamics in the concept space will thus provide a direct lens into the model's capabilities as they are acquired.

We also note that the definition above is not dependent on the precise manner via which the model is prompted to elicit an output, i.e., it need not be the case that the conditioning information $h$ that is used for training the model is used to evaluate the model capability. In fact, in our experiments, we will try alternative protocols such as intervening on the latent representations to show that substantially before the model can be prompted using $h$ to generate samples from an OOD concept class, it can generate samples from said class via such latent interventions. To this end, we also define a measure that assesses how much learning signal the data provides towards disentanglement of a concept, and hence learning of a capability to manipulate it.

**Definition 3.** *(Concept Signal.)* *The concept signal $\sigma_i$ for a concept $z_i$ measures the sensitivity of the data-generating process to change in the value of a concept variable, i.e., $\sigma_i := |\partial \mathcal{G}(z) / \partial z_i|$.*

Intuitively, if the training objective is factorized at the granularity of concepts, concept signal indicates how much the model would benefit from learning about a concept. For example, consider a diffusion model trained using the MSE loss with conditioning $h := z$ to predict the noise added to an image $\mathcal{G}(z)$. $\sigma_i$ is then merely the component of the loss representing how much change in conditioning $h$ yields changes in concept $z_i$, as it is represented in the image. Concept signal will thus serve as a crucial knob in our experiments to analyze learning dynamics in concept space and the order in which concepts are learned.

### 3.1 Experimental and Evaluation Setup

Before proceeding further, we discuss our experimental setup that concretely instantiates the formalization above. Our data-generating process is motivated by prior work on disentanglement [54–60] and involves concept classes defined by three concepts, each with two values: `color` = {red, blue}, `size` = {large, small}, and `shape`={circle, triangle}. In Sec. 4.1 and Sec. 4.4, we use two attributes: `color` (with red labeled as 0 and blue as 1) and `size` (large labeled as 0 and small as 1). We generate a total of 2048 images for each class, with objects randomly positioned within each image. We train models on classes 00 (`large red circles`), 01 (`large blue circles`), and 10 (`small red circles`), shown as blue nodes in Fig. 2, and test using class 11 (`small blue circles`), shown as pink nodes, to evaluate a model's ability to manipulate concepts and generalize OOD (see App. B.1 for further details). In Sec. 5, we will restrict to two attributes, `shape`={circle, triangle} and `color` = {red, blue}, and study the effect of noisy conditioning, i.e., what happens when concepts are correlated in the conditioning information $h$ due to some non-linear mixing

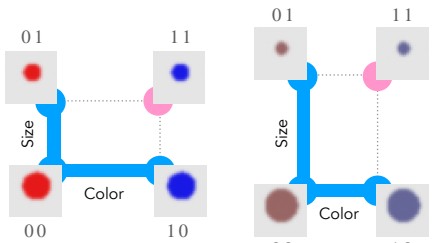

Figure 2: **Concept spaces with different concept values see different concept signal.** Consider a concept space comprised of concepts `size` and `color`. (Left) The `color` separation between the classes is stronger than the `size` separation, resulting in a stronger concept signal in the `color` dimension. (right) The `size` separation between the classes is stronger, thus resulting in a stronger signal for `size`.

function. For all experiments, we use a variational diffusion model [90] to generate $3 \times 32 \times 32$ images conditioned on vectors $h$ (see App. B.2 for further training details).

**Evaluation Metric.** To assess whether a generated image matches the desired concept class without human intervention, we follow literature on disentangled representation learning [23, 30, 54, 55, 91–94] and train classifier probes for individual concepts using the diffusion model's training data. The probe architecture involves a U-Net [95] followed by an average pooling layer and $n$ MLP classification heads for the $n$ concept variables. See App. B.2 for further details.

## 4 Learning Dynamics in Concept Space

### 4.1 Concept Signal Determines Learning Speed

We first demonstrate the utility of concept signal as a tool to gauge at what rate the model learns a concept and the capability to manipulate it. To this end, we develop controlled variants of our data-generating process by changing the level of concept signal of each concept and train diffusion models conditioned with the latent concept vector $z$ on them. We primarily focus on concepts `color` and `size`, altering their concept signal by respectively adjusting the RGB contrast between `red` and `blue` and the size difference between `large` and `small` objects (see App. B.1 for details). We define the speed of learning each concept as inverse of the number of gradient steps required to reach 80% accuracy for class `11`, i.e., the OOD class that requires learning the capability to manipulate concepts as seen during training. Results comparing different concept signals are shown in Fig. 3. For both `color` and `size`, we observe that *concept signal dictates the speed at which individual concepts are learned*. We also find that when different concepts have varying strengths of concept signals, this leads to differences in the learning speed for each concept.

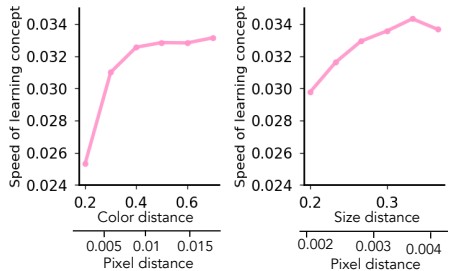

Figure 3: **Concept signal determines learning speed.** The speed of concept learning as an inverse of the time in gradient steps when the separation in color (left) and size (right) between different classes increases. Concept learning is faster when pixel differences among concept class and hence concepts are larger.

### 4.2 Concept Signal Governs Generalization Dynamics

We next examine the model's learning dynamics in concept space under various levels of concept signal for the concepts `color` and `size`. For completeness, we evaluate a model's ability to generalize

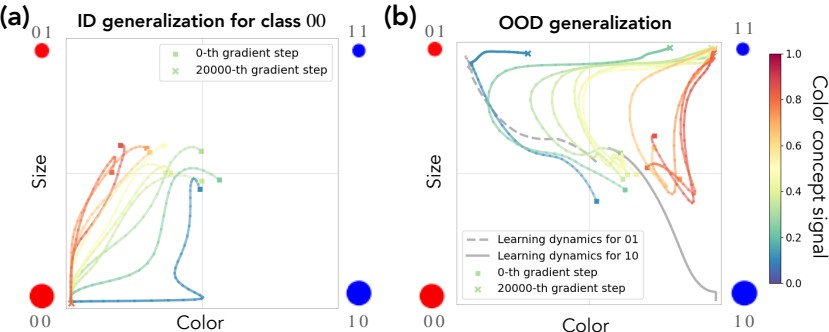

Figure 4: **Concept signal governs generalization dynamics.** (a) Learning dynamics in the concept space for in-distribution concept class `00` (bottom left). (b) Learning dynamics for out-of-distribution (OOD) concept class `11` (top right). We plot the accuracy for `color` on the x-axis and `size` on the y-axis. The [0,1) normalized `color` concept signal level is color coded. Two trajectories for `01` and `10` are shown to illustrate *concept memorization*. See App. D.3 for uncertainties.

both in-distribution (ID) and OOD, but only the latter is deemed learning of a capability to manipulate a concept. Results are shown in Fig. 4, with panel (a) showing dynamics for learning to generate samples from ID class 00 and panel (b) showing dynamics for OOD class 11. Results on OOD generalization show *concept memorization*, which we define as the phenomenon where *the model's generations on an OOD conditioning $h$ are biased towards the training class that helps define the strongest concept signal*. For example, for the unseen conditioning 11, the generations are more alike 01 when the concept signal is stronger for size (e.g., blue curve in Fig. 4(b)) and alike 10 when the signal is stronger for color (e.g., red curve). Interestingly, we observe that for settings with high imbalance in concept signals, e.g., the blue curve in Fig. 4 (b), the endpoint of *concept memorization* is very biased towards one training class, here 01, delaying its out-of-distribution (OOD) generalization. For the learning trajectories of all classes, see Fig. 15 in App. D.2.

Broadly, our results imply that an early stopped text-to-image model can witness *concept memorization* and hence simply associate an unseen conditioning to the nearest concept class when asked to generate OOD samples (see Kang et al. [96] for a similar result in LLMs). However, given sufficient time, the model will disentangle concepts underlying the data-generating process and learn to generate entirely novel, OOD samples. For futher evidence in this vein, we also confirm our results across more general scenarios, including with the real-world CelebA dataset (App. D.4) and using three concept variables: color, background color, and size (App. D.5).

### 4.3 Towards a Landscape Theory of Learning Dynamics

Fig. 4 indicates models undergo phases of understanding of concepts at different stages of training. In fact, an intriguing property of trajectories shown in Fig. 4 (b) is that there is a sudden turn in the learning dynamics from *concept memorization* to *OOD generalization* (e.g., see the top-left quadrant in Fig. 4 (b)). To investigate this further, we propose a minimal toy model that captures the geometry of model's learning trajectories shown in that figure. Specifically, we use the following dynamics equation, $d(t) \coloneqq \tilde{z} + (\hat{z} - \tilde{z}) \cdot \frac{1}{1+e^{-(t-\hat{t})}}$, where $\hat{z}$ is the target point we want to get to and $\tilde{z}$ is the initial, "biased" target. For example, consider the case with color and size concepts in Fig. 4 (b). The model's generated samples are more alike class 01 and biased towards $(\tilde{z}_1, \tilde{z}_2) = (0, 1)$ when the size concept signal $\sigma_2$ is stronger than $\sigma_1$; and to 10, $(\tilde{z}_1, \tilde{z}_2) = (1, 0)$ when the color

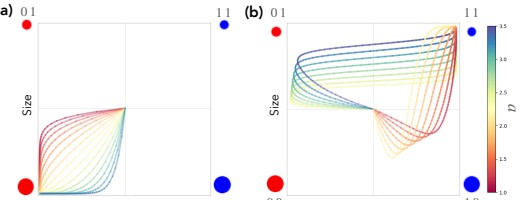

Figure 5: **A Phenomenological Model of Learning Dynamics in Concept Space.** Using Eq. 1, we simulate the learning trajectory for concept class 00 in panel (a) and the OOD class 11 in panel (b). Initially, target values are set at $(0, 1)$ or $(1, 0)$ based on the concept signal strengths for color or size, respectively. As the model progressively learns each concept, the target values gradually shift towards $(1, 1)$. This simple toy model accurately reproduces the observed curves in Fig. 3(c), which arise from *concept memorization*.

concept signal $\sigma_1$ is stronger than $\sigma_2$. We define $(\hat{z}_1, \hat{z}_2)$ as the target values or directions we want the learning to head towards (e.g., $(1, 1)$ for OOD generalization). Based on this framework, we can derive the following energy function.

$$\frac{d\mathbf{z}}{dt} = -\nabla_{\mathbf{z}}L, \; L(z_1, z_2) = \begin{cases} \frac{1}{2a}\left(d(t - \hat{t}_1) - z_1\right)^2 + \frac{a}{2}(1 - z_2)^2 & \text{if } \sigma_1 > \sigma_2, \\ \frac{1}{2a}(1 - z_1)^2 + \frac{a}{2}\left(d(t - \hat{t}_2) - z_2\right)^2 & \text{otherwise.} \end{cases} \quad (1)$$

Here, $a$ is determined by the difference $|\sigma_1 - \sigma_2|$ and $\hat{t}_1$ and $\hat{t}_2$ denote the times when the model learns concepts $z_1$ and $z_2$, respectively. Fig. 5 illustrates the simulated trajectories for classes 00 and 11, based on Eq. 1. Panels (a) and (b) correspond to classes 00 and 11, respectively. We define the actual target points $(\hat{z}_1, \hat{z}_2)$ as $(0, 0)$ for class 00 and $(1, 1)$ for class 11. For the initial targets $(\tilde{z}_1, \tilde{z}_2)$, we set both values to $(0, 0)$ for class 00. For class 11, the targets are set to $(1, 0)$ when $\sigma_1 > \sigma_2$ and to $(0, 1)$ when $\sigma_1 < \sigma_2$. *We find our toy model effectively captures the actual learning dynamics for both in-distribution (Fig. 3(b)) and out-of-distribution (OOD) concept classes (Fig. 3(c)).* Notably, our simulation accurately replicates the two types of curves: clockwise (blue trajectory in Fig. 3(b)) and counterclockwise (red trajectory).

An important conclusion that follows from the results above is that the network's learning dynamics can be precisely decomposed into two stages, hence yielding the sudden turns seen in trajectories in

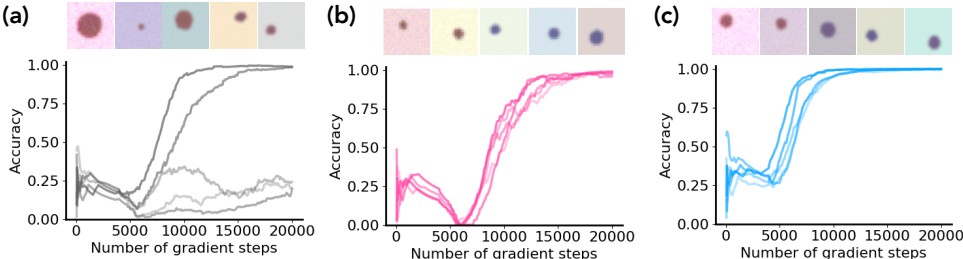

Figure 6: **Emergence of hidden capabilities.** We plot accuracy as a function of gradient steps for five different runs, using three different protocols for prompting the model to generate outputs for OOD concept classes. (a) The baseline, naive prompting protocol; (b) linear latent intervention, applied in the activation space; and (c) overprompting, akin to an intervention on the input space.

Fig. 4. *We hypothesize there is a phase change underlying this decomposition and the model acquires the capability to alter concepts at this point of phase change.* We investigate this next.

### 4.4 Sudden Transitions in Concept Learning Dynamics

The concept space visualization of learning dynamics observed in Fig. 4 (b) and our toy models analysis in Fig. 5 indicate that there exists a phase in which the model departs from concept memorization and disentangles each of the concepts, but still produces incorrect images. We claim that at the point of departure, the model has in fact already disentangled concepts underlying the data-generating process and acquired the relevant capabilities to manipulate them, hence yielding a change of direction in its learning trajectory. However, naive input prompting is insufficient to elicit these capabilities and generate samples from OOD classes, giving the impression the model is not yet "competent". This then leads to the second phase in the learning dynamics, wherein an alignment between the input space and underlying concept representations is learned. We take the model corresponding to the second from left curve (the green curve) in Fig. 4 (b) to investigate this claim in detail. Specifically, given intermittent checkpoints along the model's learning trajectory, we use the following two protocols for prompting the model to produce images corresponding to the class 11 (`blue`, `small`). See App. C for further details.

1. **Activation Space: Linear Latent Intervention.** Given conditioning vectors $h$, during inference we add or subtract components that correspond to specific concepts (e.g., $h_{\texttt{blue}}$).

2. **Input Space: Overprompting.** We simply enhance the color conditioning to values of higher magnitude, e.g. $(r,\ g,\ b) = (0.4, 0.4, 0.6)$ to $(0.3, 0.3, 0.7)$.

Fig. 6 shows the accuracy for five independent runs under: (a) naive input prompting, (b) linear latent interventions, and (c) overprompting. In Fig. 6 (a), we observe that some runs can produce samples from the target concept class (`blue`, `small`) with $\sim 100\%$ accuracy after around 8,000 gradient steps, while other runs fail to do so. However, in Fig. 6 (b, c), we find alternative protocols for prompting the model can *consistently* elicit the desired outputs much earlier than input prompting, e.g., at around as early as 6,000 gradient steps. *This indicates the model does possess the capability to alter concepts and generalize OOD!* Furthermore, we note that different protocols enable elicitation of the capability *at approximately the same number of gradient steps, irrespective of the seed, and that this is precisely the point of sudden turn in the learning dynamics in Fig. 4*! Interestingly, experiments with Classifier Free Guidance (CFG) [97] show that CFG only becomes effective after this transition (Fig. 21).

We further explore the second phase of learning in Appendix D.7 by patching the embedding module used for processing the conditioning information from the final checkpoint to an intermediate one. Our results show that when the final checkpoint does enable use of naive input prompting for eliciting a capability, the embedding module can be patched to an intermediate checkpoint and we can retrieve the desired output at approximately the same time that alternative prompting protocols start to work well. This suggests the second phase of learning primarily involves aligning the input space to intermediate representations that enable eliciting the model capabilities. Overall, our results above yield the following hypothesis.

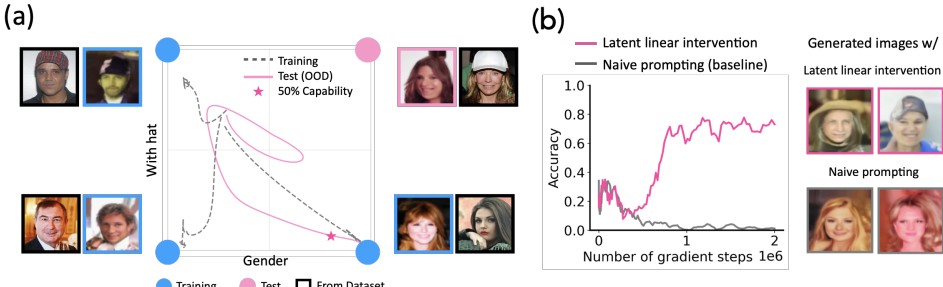

Figure 7: **Validating our Findings on CelebA.** (a) Concept space dynamics of generated images concepts `Gender` and `With Hat`. We find the generalization class (`Female`, `With Hat`) ends up near (`Female`, `No Hat`). (b) Compositional Generalization Accuracy when performing latent interventions vs. naive prompting. This rise of accuracy near $5 \times 10^5$ gradient steps clearly demonstrates that the model is *capable* of generalizing out-of-distribution, but is not *performant* when evaluated via naive prompting.

**Hypothesis 1.** *(**Emergence of Hidden Capabilities.**) Generative models possess hidden capabilities that are learned suddenly and consistently during training, but naive input prompting may not elicit these capabilities, hence hiding how "competent" the model actually is.*

### 4.5   Additional Results on Realistic Data

To provide further support for our hypothesis, we provide results in a more realistic and naturalistic data. Specifically, we use the CelebA dataset [98], which contains fine-grained attributes corresponding to concepts like `Gender`, `With Hat`, and `Smiling`, and analyze two settings.

- **Using concepts** `Gender` **and** `Smiling`. Results are in Fig. 18, where we find that concept learning dynamics discovered in the relatively simple setting in prior sections generalizes. We see that there is first a phase of concept memorization, wherein the model learns to generate images of (`Female`, `Smiling`). However, as training proceeds, there is a sudden turn in the learning dynamics and the model learns to generalize out-of-distribution.

- **Using concepts** `Gender` **and** `With Hat`. In this setting, given a compute budget of 2M gradient steps, we find that the model seem to never learn to generalize out-of-distribution (see Fig. 7). However, when we perform latent interventions on the model's representations, we are able to force the model to produce images from the class (`Female`, `With Hat`). These results reiterate that the model is indeed *capable* of generalizing out-of-distribution, but is not *performant* when naively prompted.

## 5   Effect of Underspecification on Learning Dynamics in Concept Space

In the results above, we use conditioning information that perfectly specifies concepts underlying the data-generating process, i.e., $h = z$. In practice, however, instructions are underspecified and one can thus expect correlations between concepts in the conditioning information extracted from those instructions [99–103]. For example, images of a `strawberry` are often correlated with the color `red` (see Fig. 9(a)). Correspondingly, unless a text-to-image model is shown explicit data stating "`red strawberry`" or images of non-red strawberries, the model's ability to disentangle the concept `color` from the concept `strawberry` may be impeded (see generations for "`yellow strawberry`" in Fig. 9). Motivated by this, we next investigate the effects of using underspecified conditioning information on a model's ability to learn concepts and capabilities to manipulate them.

**Experimental setup.** The data generation and evaluation process follows the protocol described in Sec. 3.1. We select

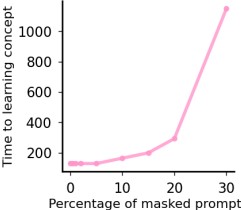

Figure 8: **Underspecification delays out-of-distribution (OOD) generalization.** The number of gradient steps required to reach accuracy above 0.8, as the percentage of masked prompts increases. A higher proportion of masked prompts slows down the speed of concept learning.

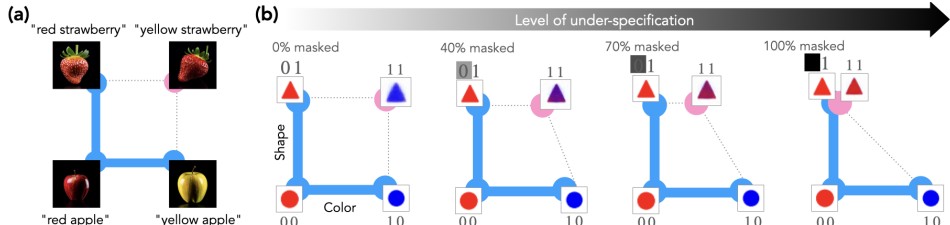

Figure 9: **Underspecification and Concept Learning.** (a) The state-of-the-art generative models [104] erroneously produces a `red strawberry` (top right corner) for the prompt "`yellow strawberry`". (b) Without underspecification in the training data, a model $F$ accurately learns the concepts of shape and color, successfully generalizes to the unseen node `blue triangle` (leftmost). As masks are applied to the word `red` for the prompt `red triangle`, concept signal for `triangle` increasingly starts to correlate with the concept `red`. This causes the output images to change from blue to purple as the level of masking increases (panels left to right). Eventually, the color dimension for `triangle` collapses, biasing the model towards generating solely `red triangles` (rightmost).

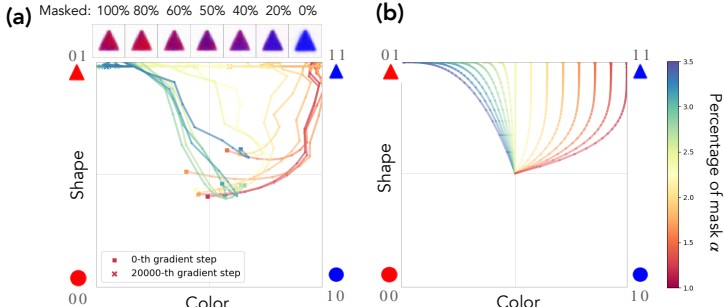

Figure 10: **Underspecification hinders out-of-distribution (OOD) generalization.** (a) The learning dynamics with varying levels of prompt masking, from 0% to 100%, and the generated images. At 0% masking (top right image), the model correctly produces an image of `blue triangle` from the prompt "`blue triangle`." As the masking increases (from right to left), the images gradually shift towards the incorrect color, `red`. (b) The simulation of the learning dynamics under underspecification in concept space based on Eq. 2. Our toy model replicates a trained network's learning dynamics.

`color` (red and blue) and `shape` (circle and triangle) as the concepts, drawing an analogy to the "`yellow strawberry`" example. To simulate underspecification, we randomly select training samples that have a specific combination of shape and color (e.g., "`red triangle`"). We then mask the token representing the color ("`red`") and train the model on three concept classes {`00`, `01`, `10`}, represented by blue nodes in Fig. 9, with some prompts masked. We test using class `11` (pink node) with no prompts masked to see if the model can generalize OOD.

**Underspecification Delays and Hinders OOD generalization.** Fig. 8 shows how underspecification (masked prompts) affects the speed of concept learning. We see that as the percentage of masked prompts increases, the speed of learning a concept decreases, suggesting that underspecification leads to slower learning of concepts. Further, Fig. 10 (a) shows models' learning dynamics in concept space at varying levels of underspecification. With 0% masking, the model accurately produces an image of `blue triangle` (see Fig. 9(b)). However, as the percentage of masking increases, the color of generated images shifts from `blue` to `purple` (middle), and finally `red`. This demonstrates that when prompts are masked, the model's understanding of shape `triangle` becomes intertwined with color `red`; even when `blue` is specified in the prompt, the dynamics are biased towards `red`.

**Toy Model of Learning Dynamics with Underspecification.** When prompts are masked (i.e., underspecification occurs), target values for the concept variables are shifted: e.g., in our setup, with no mask applied, the target directions for the class `11` is $(1, 1)$. When the word "`red`" in "`red triangle`" is fully masked, the target shifts to $(0, 1)$. Assuming this shift is linear with respect to the percentage of masked prompts $\alpha$, we can derive the following energy function.

$$\frac{d\mathbf{z}}{dt} = -\nabla_{\mathbf{z}} L, \ L(z_1, z_2) = \big((1 - s\alpha) - z_1\big)^2 + \big(1 - z_2\big)^2. \tag{2}$$

In the above, parameter $s$ represents the impact of underspecification. Fig. 10 (b) shows the simulation of model behavior in the concept space according to Eq. 2 ($s = 0.01$). As the masking level increases, the target directions shift from $z_2 = 1$ in the top right corner to $z_2 = 0$ in the top left corner. Our simulated dynamics thus match well with the model's learning dynamics shown in Fig. 10 (a).

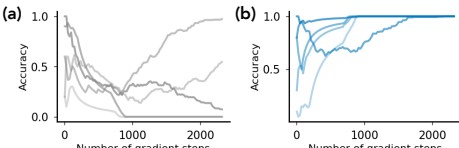

**Influence of Underspecification on Emergence of Hidden Capabilities.** Following Sec. 4.4, we also explore whether it is possible to elicit the desired outputs from a model trained with underspecified data. Fig. 11 shows the accuracy results over five runs (a) without using any prompting method, and (b) using over-prompting. In both scenarios, the percentage of masking is set at 50%. Results clearly demonstrate that with over-prompting, the model achieves 100% accuracy after approximately 1,000 gradient steps, whereas without over-prompting, it fails in three out of five runs even after 2,000 gradient steps. These findings confirm that capability can develop prior to observable behavior, even in cases of underspecification.

Figure 11: **Underspecification and Hidden Capabilities.** We use the over-prompting protocol from Sec. 4.4 to assess if the capabilities to enable OOD generalization are learned before naive input prompting. (a) Accuracy for OOD generalization across five different seed runs under naive prompting; (b) Accuracy under over-prompting, with a fixed masking percentage of 50%.

## 6 Discussion

**Why study the concept space?** One might ask why concept space could be useful beyond synthetic datasets. In Fig. 14, we show the loss, accuracy, and training trajectory in concept space of a model from Fig. 6. The moment during training when the model acquires the capability to manipulate `size` and `color` independently is not evident from either the loss or accuracy curve. However, in concept space, one can directly see the divergence of the trajectory from *concept memorization*. Benchmarking generative models is a challenging task, still often involving humans in the loop [105, 106]. Our concept space framework suggests that benchmarking core generalization capabilities can potentially be reduced to monitoring the learning trajectory in concept space. Moreover, this information can be used to develop better training schemes as discussed in Ren et al. [107].

**Concept learning vs. grokking** We make a distinction between the potentially delayed elicitation of capabilities in our model versus grokking [108]. The phenomenology of delayed increase in performance on the test set, as seen in Fig. 14, is shared. However, we deal with out-of-distribution evaluations, which is different from setups like modular arithmetic or polynomial regression in which grokking is usually studied [108–112]. Secondly, research on grokking using hidden progress measures indicates that the model gradually builds representations toward an ideal one [111, 113]; however, our results find that even at the level of latent representations, there is a sudden emergence whereby the model learns the capability to manipulate concepts underlying the data-generating process and generalize OOD.

**Is concept learning a phase transition?** In Sec. 4.4, we have demonstrated that before the capability to manipulate a concept is learned, the desired outputs cannot be generated regardless of the protocol used to prompt the model. Moreover, we showed that different protocols yield the desired outputs at the exact same time, which is also highly independent of model initialization (Fig. 6, Fig. 17). We thus hypothesize that: *Concept learning is a well-controlled phase transition in model capability. However, the ability to elicit this capability through predefined, single-input prompting can be delayed arbitrarily, depending on factors like the strength of the concept signal.*

**Limitations** One limitation of our work is that the compositional setup used in this study is rather simple. Real world concepts are often hierarchical [114, 115] or relational [116]. We also make assumptions that concepts are linearly embedded in the vector space $\mathcal{Z}$ in Sec. 3. However, this assumption seems to be at least partially justified as seen in Sec. D.8. Another limitation is that the findings are mainly drawn from synthetic data. While we show that our findings generalize to realistic data to some extent, as seen in Sec 4.5, Fig. 18 and Fig. 7, further studies are needed to investigate how our findings apply to real data in general.

## Acknowledgments and Disclosure of Funding

CFP and HT gratefully acknowledges the support of Aravinthan D.T. Samuel. CFP, MO, ESL and HT are supported by NTT Research under the CBS-NTT Physics of Intelligence program. ESL's time at University of Michigan was partially supported by the National Science Foundation (IIS-2008151 and CNS-2211509). The computations in this paper were run on the FASRC cluster supported by the FAS Division of Science Research Computing Group at Harvard University. The authors thanks Zechen Zhang, Helena Casademunt, Carolina Cuesta-Lazaro, Shivam Raval, Yongyi Yang and Itamar Pres for useful discussions.

## Contributions

Motivated by prior work from MO, ESL and HT [30], CFP and HT started investigating the order in which concepts are learned by a diffusion model, leading to CFP identifying the notion of concept signal and HT proposing to study the learning dynamics in concept space. This kickstarted the project. ESL and HT defined the formal setup (Sec. 3). ESL hypothesized the point of sudden turn marks the emergence of hidden abilities, hence defining the project narrative around competence versus performance. AL and CFP led the experimental verification of this hypothesis. AL led identification of linear representation of concepts and performed the latent and MLP intervention, while CFP contributed to the input space one. MO and HT led formulation of the landscape theory (Sec. 4.3), with inputs from CFP. MO led the underspecification picture of learning dynamics, with inputs from ESL and HT (Sec. 5). Paper writing was led by MO and ESL, with inputs from all authors. Visualizations were mainly led by MO, with inputs from HT and CFP. Experiments on CelebA learning dynamics were conducted by CFP in discussions with MO and corresponding latent intervention were led by AL.

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

## A Schematic for Concept Space

Fig. 12 illustrates the concept space framework discussed in Sec. 3.

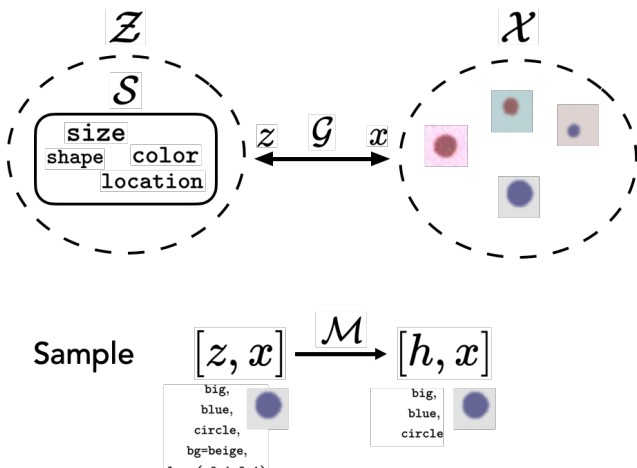

Figure 12: **The concept space framework.** $\mathcal{G}$ is an invertible data-generating that maps process maps sampled vectors $z \sim P(\mathcal{Z})$ (where $\mathcal{Z} \subset \mathbb{R}^d$) to an observation $x \in \mathbb{R}^n$ (an image in this work). The concept space, $\mathcal{S} := \{z | z \sim P(\mathcal{Z})\}$, is defined as a space of all possible concept vectors generated from a compositional prior $P(\mathcal{Z})$. The mixing function $\mathcal{M}$ masks some concept variables, the masked concept variables are thus underspecified.

## B Experimental Details

### B.1 Synthetic data

We design a data generation process (DGP) compatible with the concept space framework introduced in Sec. 3. Our full DGP has six concepts: `shape`={circle, triangle}, `x coordinate`$\in \mathbb{R}$, `y coordinate`$\in \mathbb{R}$, `color`={red, blue}, `size`={big, small}, and `background color`={bright, dark}. We generally explore only a subset of these at a given time in our experiments.

In Sec. 4, we fix `shape`=circle and `x coordinate, y coordinate, background color` are masked out. Thus the conditioning vector $h$ only specifies `color`={red, blue} and `size`={big, small}. We sample both of these concept variables from a mixture of two uniform distributions, one component for each class (big, small) and (red, blue). Each value in each dimension is sampled from $\mathcal{U}(m^i - s, m^i + s)$, where $m^i$ is the class dependent mean. For example, `color`=red could be sampled from $\mathcal{U}(0.7, 0.9) \times \mathcal{U}(0.1, 0.3) \times \mathcal{U}(0.1, 0.3)$, where $m^0 = (0.8, 0.2, 0.2)$ For brevity, we name the four resulting classes "00", "01", "10", and "11", where the class "11" is kept as the unseen target to evaluate out-of-distribution (OOD) generalization. For each training run, the DGP was initialized with a set random seed and 2048 images were generated in each class.

In Sec. 5, we used two concept variables, `shape` and `size`. The training dataset included the classes "00", "01", and "10". For evaluation, we used the class "11". For each class, we generated a total of 1,000 images, each featuring objects of varying positions and sizes to ensure variability in our dataset.

In App. D.5, we add the concept variable `background color`$\in \mathbb{R}^3$ to have a three dimensional concept space of (`color`, `size`, `background color`). In this case our training data is composed of the classes ("000", "001", "010", "100") and the test classes are ("011", "101", "110", "111")

Our DGP is resolution agnostic, but we work with square images of 32x32 for fast experiments.

**Adjusting concept signal** To adjust concept signals, we vary the class dependent mean of the two components of the mixture distributions. For example a strong color concept signal is achieved by drawing `color`=red from the mean $m^{red} = (0.9, 0.1, 0.1)$ and `color`=blue from the mean $m^{blue} = (0.1, 0.1, 0.9)$ whereas a weak color concept signal can be drawn using $m^{red} = (0.6, 0.4, 0.4)$ and $m^{blue} = (0.4, 0.4, 0.6)$. We scale the standard deviation of each component by the same ratio the

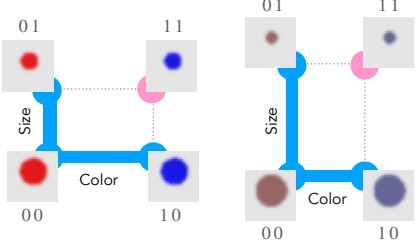

Figure 13: **Different distributions in concept values result in different concept signal.** (Left) The `color` separation between the classes is stronger than the `size` separations resulting in a stronger concept signal in the `color` dimension. (right) The `size` separation between the classes is stronger, thus resulting in a stronger concept signal in `size`

difference between the class means has been scaled. This latter is important to avoid the model's natural interpolation or extrapolation abilities from confounding with the generalization dynamics.

Fig. 13 illustrates two datasets with varying concept signals. Fig. 13 (left) illustrates a case where the `color` concept signal is stronger than the `size` concept signal, while Fig. 13 (right) illustrates the inverse case where the `size` concept signal is stronger.

## B.2 Model Details

**Model Architecture** We used a network architecture similar to the U-Net [95] used in Variational Diffusion Models [90]. We used the implementation publicly available in [117]. The architecture has 2 ResNet [118] blocks before each downsampling layer and a has self-attention layer in the bottleneck layer of the network. The U-Net has 64, 128, 256 channels at each resolution levels, an has a LayerNorm [119] and a GELU [120] activation. We used a sinusoidal timestep embedding [121] of 64 dimensions followed by a 2-layer MLP with hidden dimension 256 and a conditioning vector embedding using a 2-layer MLP with hidden dimensions 256 independent in every residual block.

**Optimizer** We use the AdamW [122] optimizer with learning rate 0.001 and weight decay 0.01 to optimize the parameters of our network. We train our networks for 20K gradient steps. We use the default values for the decay rates: $\beta_1 = 0.9$, $\beta_2 = 0.999$.

**Classifier Free Guidance** At training time we drop the conditioning to a $-1.0$ filled null vector $\phi$ with probability 0.2. This unconventional choice was made instead of the dropping it to the null vector since the null vector was near the interpolation limit of our model in the color subspace of the conditioning. The $w_{cfg}$ parameter is used to estimate the noise in the diffusion process by: $\hat{\epsilon}_{t-1} = f_\theta(x_t|\phi) + w_{cfg} * (f_\theta(x_t|h) - f_\theta(x_t|\phi))$, where $\theta$ denotes parameters of the network $f$.

**Diffusion Process Hyperparameters** We used the continuous time variational diffusion model [90] in order to (i) keep the sampling step parameter T an inference time hyperparameter and (ii) to allow the model to the adjust its optimal noise schedule for these images, especially since our synthetic images are expected to be different in SNR from natural images. In particular, we initialize our network with $\gamma_i = -5.0$ and $\gamma_f = 10.0$ (see [90] for the definition of $\gamma$) and use a learned linear schedule of $\gamma(t)$. We use a reconstruction loss corresponding to a negative log likelihood from a standard normal with $\sigma = 10^{-3}$ centered at the data for the first step of the diffusion process.

**Evaluation probe Details.** We used a U-Net [95] backbone with 64 output channels followed by a max pooling layer and $n \times$ 1-layer MLP classifier for each of the $n$ classes to estimate each concept of an image independently. We sample from the same data distribution but with maximal data diversity, i.e., with $s_i$ values in App. B.1 maximized within the range allowing perfect classification (no overlap of $z$ between classes). We sample 4096 images per class from the DGP and train the classifier for 10K gradient steps with AdamW [122] and achieve a 100% accuracy on the held out test set. At evaluation phase, we average the classifier softmax output over 5 data generation / model initialization seeds and 32 inference samples to construct the concept space representation of the generations.

**Hyperparameter Search** We conducted a hyperparameter search, testing batch sizes from 32 to 256, number of channels per layer from 64 to 512, learning rates between $10^{-4}$ and $10^{-3}$, the number of steps in the diffusion process from 100 to 400, weight decay between $3 \times 10^{-3}$ and $5 \times 10^{-2}$,

and model weight initialization scale between $\mathcal{N}(0, 0.003)$ and $\mathcal{N}(0, 1)$. We also tried the Adam optimizer [123] with $\beta_1 = 0.9$, $\beta_2 = 0.99$, and weight decay of $10^{-5}$. No qualitative change were found in our results.

**Computational Details.** We implement our models in PyTorch [124]. The diffusion model was trained on four Nvidia A100 GPUs and 64 CPUs for the data generating process. A standard model run (e.g., in Sec. 4.2) took $\sim 20$ minutes on *a single* NVIDIA A100 40GB GPU. The CelebA runs took $\sim 24$ hours on the same GPU. The full research required roughly 500 GPU hours, while the experiments in the paper would require roughly 100 GPU hours.

### B.3    Code Availability

Our code is available at https://github.com/cfpark00/concept-learning.

## C    Details on Alternative Protocols for Eliciting Model Capabilities

**Input Space: Overprompting.**    Our model is trained on a distribution of conditioning vectors centered around a class dependent mean: $p^i(z_j) = \mathcal{U}(m_j^i - s_j, m_j^i + s_j)$ for each class $i$. We prompt the model with conditioning vectors extrapolated in the direction $\vec{m}_1 - \vec{m}_0$. For instance, assuming the red conditioning was $(0.6, 0.4, 0.4)$ and the blue conditioning was $(0.4, 0.4, 0.6)$, we "prompt" the model with $(0.2, 0.2, 0.8)$ for blue. In practice, we use 5 conditionings $(0.4, 0.4, 0.6)$, $(0.35, 0.35, 0.65)$, $(0.25, 0.25, 0.75)$, $(0.15, 0.15, 0.85)$, $(0.05, 0.05, 0.95)$, and report the maximum joint accuracy.

**Activation Space: Linear Latent Intervention.**    We demonstrate the ability to compose capabilities by manipulating the conditional vectors $\vec{h}$. Namely, we create a condition vector $\vec{h}_i$ for a specific concept $i$ by specifying a concept of interest (e.g., $\vec{h}_{\mathtt{blue}} = M(z_{\mathtt{blue}})$). During the forward pass, given $\vec{h}$, we can compute the component of each concept in $\vec{h}$ by projecting onto a specific concept-condition vector ($\vec{h}_{\mathtt{blue}}$). We can then enhance or reduce the component of each concept by scaling each of these projected components. In practice, we perform the following operation: $\vec{h}' = \vec{h} + \alpha \vec{h}_{\mathtt{blue}} - \beta \vec{h}_{\mathtt{large}}$, where $\alpha, \beta$ are hyperparameters. We sweep over $[0.1, 1, 2, 4]$ for $\alpha$ and $[0.1, 0.25, 0.5, 1]$ for $\beta$.

$\vec{h}_{\mathtt{blue}}$ is constructed by first deriving a blue direction in condition embedding space ($\vec{h}_{5,5,95}$) by embedding a concept vector ($\vec{z}_{5,5,95}$) where its RGB components are set to $(0.05, 0.05, 0.95)$: $\vec{h}_{5,5,95} = M(\vec{z}_{5,5,95})$. We then project $\vec{h}$ onto this direction to derive $\vec{h}_{\mathtt{blue}}$. $\vec{h}_{\mathtt{large}}$ is generated similarly using $\vec{z}_{\mathtt{size}=0.7}$. The model then generates an image conditioned on $\vec{h}'$.

## D    Additional Results

### D.1    Loss and Accuracy versus Concept Space (Fig. 14)

In Fig. 14, we plot loss, accuracy, and OOD generalization trajectory in concept space of the models for a training run in Fig. 6. The time at which the model acquires a capability to manipulate the concept `color` is not evident from the loss or accuracy curves; however, in concept space, one can identify when the trajectory deviates from *concept memorization* and thus when the compositional ability has emerged. Benchmarking generative models is a challenging task, still often involving humans in the loop [105, 106]. Our concept space framework suggests that benchmarking out-of-distribution (OOD) generalization can potentially be reduced to monitoring the learning trajectory in concept space.

### D.2    Additional trajectories of learning dynamics (Fig. 15)

Here we show all concept space trajectories for the experiments mentioned in Fig. 4 (a,b), for all classes and `color` concept signal levels. We find asymmetric behavior for the `01` class and the `10` class when adjusting the `color` concept signal level. The dynamics of the generations in the training set matches our intuitions of concept signal as discussed in the main text. At low `color` concept

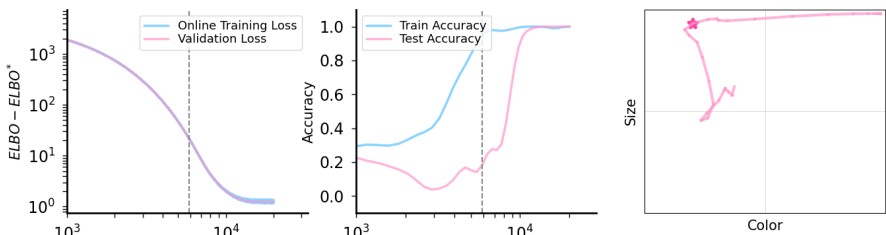

Figure 14: **Loss vs. accuracy vs. concept space.** (left) The training and validation loss on ID classes during training. (center) The accuracy on the training classes and the test (OOD generalization) class. (right) Concept space trajectory of the generalization class. Loss and accuracy do not always intuitively reflect what capabilities the model has acquired during training. However, as one can see in the rightmost panel, the point of sudden turn in concept space corresponds to when the capability has emerged, i.e., the moment when well defined prompting protocols can elicit the desired output from the model, which we indicate using the pink star (50% capability).

signal, we observe that the dynamics fit `size` first for both `00` and `10`, and afterwards find their correct colors.

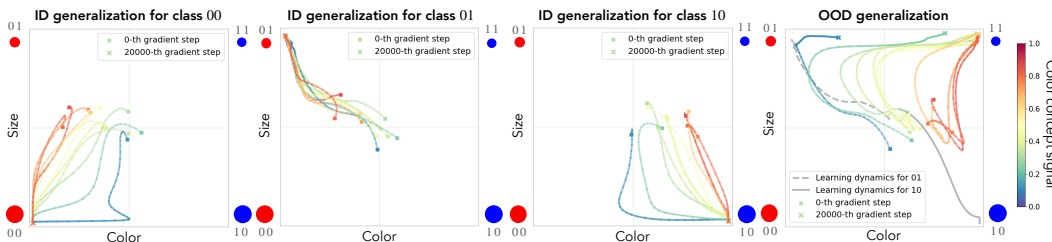

Figure 15: **Concept space dynamics for all classes (`00`, `01`, `10`, `11`).** The experiment is identical as in Fig. 4. The [0,1] normalized `color` concept signal is color coded in every trajectory. Two training data trajectories are shown in gray in the last panel to illustrate *concept memorization*.

### D.3 Error Quantification (Fig. 16, Fig. 17)

We show the standard error of the mean (s.e.m.) of concept space trajectories of Fig. 4 and Fig. 15 in Fig. 16 (a). Fig. 16 (b) illustrates the 50% capability point across 4 different model initialization seeds. Fig. 16 (c) illustrates the 50% capability point across different color concept signal level. Overall, our results are consistent over random seeds and concept signal levels.

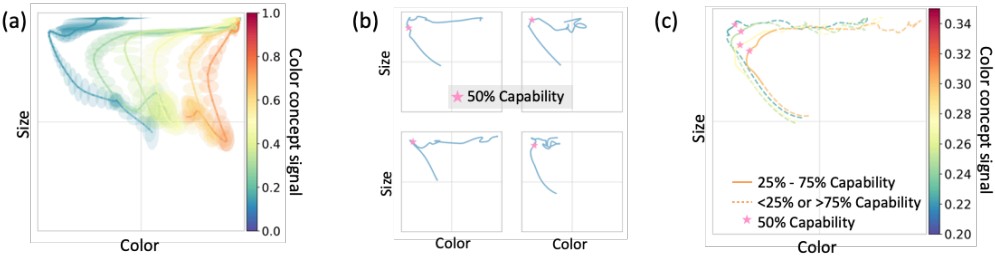

Figure 16: **Error quantification on concept space trajectories.** (a) Concept space trajectories and their standard error of the mean (n=5) for different concept signal levels. (b) Concept space trajectories for 4 different seeds. The pink stars denote the point of 50% capability. (c) Concept space trajectories for different color concept signal level. The stars again denote the 50% capability point and the 25% − 75% capability regions are in solid lines.

We show the standard deviation of compositional generalization accuracy for different prompting methods in Fig. 17. We find that the standard deviation is large for the naive prompting method, while linear latent interpolation and overprompting extract the hidden capability *robustly across seeds*, resulting in a small standard deviation.

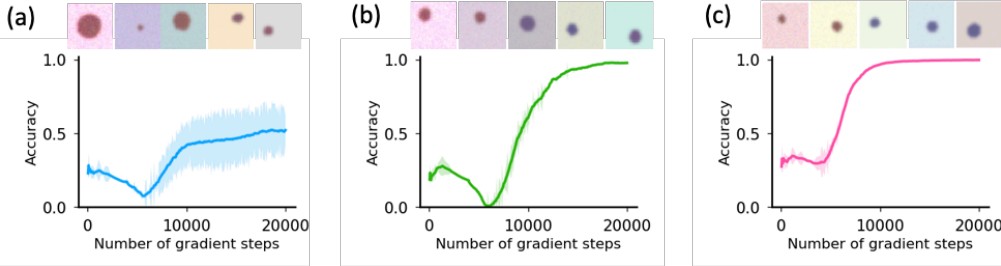

Figure 17: **Standard deviation of compositional generalization accuracy.** We show the compositional generalization accuracy and shade the $1\sigma$ region for (a) Naive prompting (b) Linear latent intervention (c) Overprompting

### D.4 Additional Experiments with real data, CelebA (Fig. 18)

To assess our findings on a more realistic dataset, we ran experiments on the CelebA [98] dataset. The experiments in Fig. 7 and Fig. 18, share the same experiment hyperparameters, described below.

We selected the `Female` and `Smiling` features as the two concept dimensions to explore. This choice was motivated by the relatively balanced number of samples in all four classes (00=(`Male`, `Not Smiling`), 01=(`Male`, `Smiling`), 10=(`Female`, `Not Smiling`), 11=(`Female`, `Smiling`)), from which we randomly sampled 30K images in each class. To construct the concept space, we trained a fully convolutional network with a average pooling layer followed by a classification MLP head to classify the two attributes with independent cross entropy losses. (See B.2). We trained the classifier with the AdamW [122] optimizer with learning rate $10^{-3}$ and weight decay $10^{-5}$ for 10K gradient steps. The classifier achieved a final accuracy of respectively $95\%$ and $97\%$ on the held out validation set, which was $10\%$ of the entire dataset.

We trained the same diffusion model (See App. B.2) from our synthetic experiments on 64x64 resized images from the classes (00, 01, 10) and assessed the out-of-distribution (OOD) generalization to 11. We used a color jitter of 0.1 for brightness, contrast and saturation and randomly flipped the images horizontally. As the class attributes are categorical, they were one hot encoded and concatenated to an input conditioning vector of 4 dimensions. The diffusion model was trained for $2 \times 10^6$ gradient steps with a batch size of 64 with the same optimizer as in the main experiments.

The concept space dynamics of generations from the in distribution conditioning and OOD conditioning are shown in Fig. 18. We find similar observations from the concept space trajectories as in our synthetic experiments. Initially, the images corresponding to the class 11 follows the concept space trajectory of 10, optimizing `Female`, which we intuitively expect to have a stronger concept signal, although it is intractable to compute since the DGP is unknown. Similar to our synthetic experiments, we see a transition in the concept space trajectory of the compositional class corresponding to the model disentangling the concepts. After this transition, concept learning begins but we observe a substantial bias towards the training distribution. We see a trade-off of moving in the right direction towards one concept degrades the other one similarly to the observations in Fig. 4 (b) for high color concept signal levels. The visual generations in Fig. 18 confirm that our findings are not merely a result of an ill-calibrated classifier model. However, in this case, we do not observe full out-of-distribution (OOD) generalization at 2M gradient steps. We expect this task of generating (`Female`, `Smiling`) is inherently harder than our synthetic setup. We note that the goal of this experiment is not to show good compositional generalization but it is more on confirming that our qualitative findings generalize to real data without touching the model or training method.

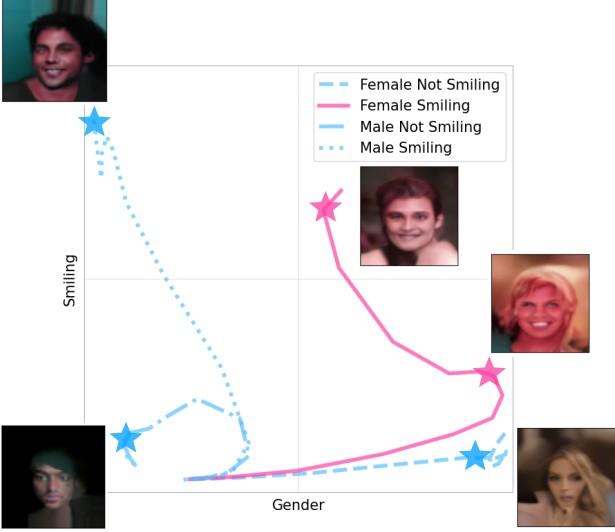

Figure 18: **Concept space dynamics with CelebA.** We train on the classes 00=(Male, NOT Smiling), 01=(Male, Smiling), 10=(Female, NOT Smiling) (plotted in blue) and test on the class 11=(Female, Smiling) (plotted in pink). We find similar observations as in Fig. 4.2, *concept memorization* and the resulting bias.

## D.5 Experiments with 3D concept space (Fig. 19, Fig. 20)

To further verify our findings in Sec. 4, we explore a setup where the concept conditioning specifies three concepts: `color`, `size` and `background color`. Fig. 19 illustrates two example scenarios in which the concept signal in `color` and `background color` are varied, respectively resulting in cases of success and failure of OOD generalization by naive prompting. The length of edges of the cuboids represent their concept signal magnitude. In Fig. 19 (a), we see that the object color has a strong concept signal and this is reflected in the concept space trajectories as this direction being split first. In this case we see, similarly to the blue generalization curve in Fig. 4 (b) and Fig. 15 (rightmost panel), a slow generalization process for the compositional class 111. Again similarly to our findings in Sec. 4.2, we observe that the 011 class initially undergoes *concept memorization* for class 010, which shares the two stronger concept signals `color` and `background color`, and shows a transition where it suddenly curves out from this trajectory. In Fig. 19 (b), we see a case where out-of-distribution (OOD) generalization did not succeed within the given 80K gradient steps. In this case, we see two classes, 011 and 111, which didn't reach the right compositional class via naive prompting. However, the concept space trajectories suggest that the *capability* to compositionally generalize `background color` has already emerged as visible from the sharp turns.

Fig. 20 quantifies the accuracy of each concept separately. Fig. 20 (a) shows that stronger concept signal accelerates the corresponding concept learning. Fig. 20 (b) shows that other concepts' learning speed can vary, albeit smoothly, when changing a single concept signal while the corresponding concept is the most affected.

## D.6 Experiments with classifier free guidance (Fig. 21)

An implication of our conclusions in Sec. 4.4 is that before the model has passed the transition point where the model has the *capability* to compose concepts, the model should not be able to generate small blue circles no matter how well "prompted". Here, instead of prompting, we explore Classifier Free Guidance (CFG) [97] to see if our findings apply to a conditional diffusion model trained with CFG. In Fig. 21, we see that even the models with CFG show this transition from *concept memorization* to out-of-distribution (OOD) generalization. In a scenario where there isn't a sharp acquisition of the capability to compositionally generalize, we would expect the sharp transition to disappear with CFG scale. However, as we observe in Fig. 21, the sharp turn in concept space still exists with CFG, perhaps even more pronounced than baseline experiments. These results suggest

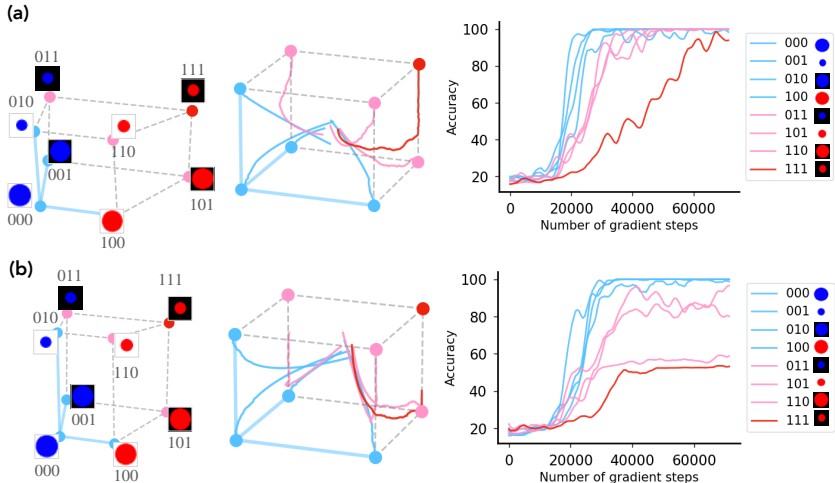

Figure 19: **Generalization dynamics in a 3 concept setting.** (a) Generalization dynamics when `color` carries the strongest concept signal. (b) Generalization dynamics when `size` carries the strongest concept signal. We observe sharp turns corresponding to learning background color, however, naive prompting cannot fully elicit compositonal generalization.

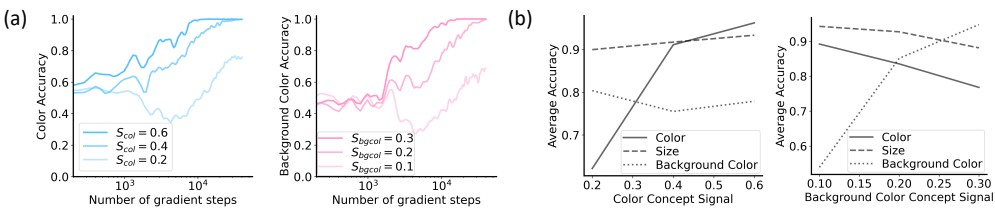

Figure 20: **Accuracy on individual concepts in a 3 concept setting.** a) We show the dynamics of `color` and `background color` accuracy during model training, depending on different concept signal levels b) We show the average accuracy until the end of training as a metric of learning speed for different concepts, as we change a single concept signal level.

that the acquisition of the compositional generalization ability is required for CFG to enhance OOD generalization.

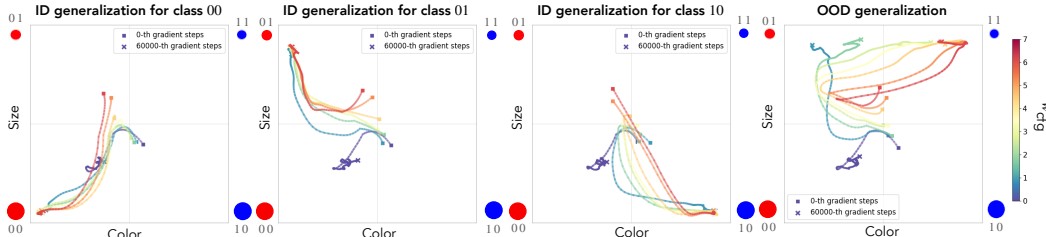

Figure 21: **Learning dynamics in Concept space for classifier-free Guidance** We show the learning dynamics in the concept space for all the classes (00, 01, 10, 11). The generalization task (rightmost) shows a sharp transition from *concept memorization* to OOD generalization independently of the classifier-free guidance scale.

### D.7 Patching the embedding module (Fig. 22)

We test an additional elicitation method where we swap the conditioning vector ($h$)'s embedding module with that of the last checkpoint. An interpretation of this method is that the embedding module disentangles the concepts, i.e., generates a representation for each concept, while the U-Net [95] then

utilizes such representations. This would imply that the U-Net [95] learns how to utilize concept representations early during training, while many more gradient steps are needed for the concepts to be disentangled when naively prompted.

We test our approach on 5 random initialization seeds. Results are shown in Fig. 22. We find that for some seeds, we are able to elicit the target behavior at around the same time in which overprompting and linear interventions also elicit the target behavior; for other seeds, this is not the case. These results demonstrates that the role of the embedding module and the U-Net *might* be separated during training for some runs without an explicit term enhancing this separation of roles.

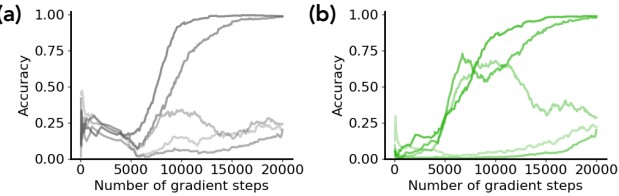

Figure 22: **Embedding patching.** We patch the embedding module (an MLP) used for transforming the conditioning information into an embedding that the model processes from the last checkpoint to intermediate checkpoints. Panel (a) shows the baseline accuracy for out-of-distribution (OOD) generalization across five different seed runs, while panel (b) shows accuracy achieved when the patched embedding module is used.

### D.8 Proof of Concept Experiments with Frontier Models (Fig. 23)

We show simple proof of concept experiments on frontier models in Fig. 23. In Fig. 23 (a), we show that CLIP [125] already embeds text/images with compositional concepts in its vector space as a cube respecting the Hamming graph of concepts. In Fig. 23 (b), we show that simple over-prompting can enhance compositional generalization ability in Stable Diffusion v1.4 [126]. These experiments show that the assumptions we made to construct the concept space framework in Sec. 3 and the elicitation method we used work at least to some extent in frontier models.

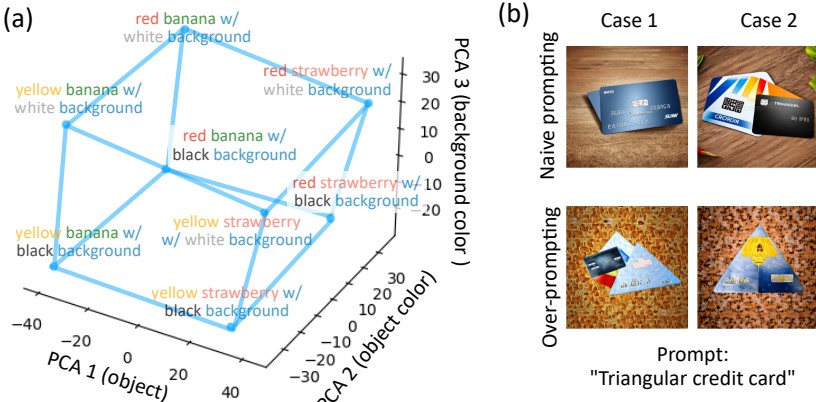

Figure 23: **Proof of concept experiments on frontier models** (a) CLIP [125] embeds compositional text/images as a concept cube in its vector space. (b) Simple over-prompting can enhance compositional generalization in Stable Diffusion v1.4 [126]

