# OpenReview forum: "Emergence of Hidden Capabilities: Exploring Learning Dynamics in Concept Space"
_NeurIPS.cc/2024/Conference — NeurIPS 2024 spotlight_

### Official Review · Reviewer_o8es · 2024-06-18

**Soundness:** 3
**Presentation:** 4
**Contribution:** 4
**Rating:** 8
**Confidence:** 4

**Summary:**

The paper introduces a novel framework for understanding concept learning in text-conditioned generative models. The experiments are carefully designed (although most of them look a bit toyish) and support the analyses well. The three key findings, i.e., concept signal levels determine the learning dynamics; capability consistently emerges before observable behaviors; under-specification harms the system, are quite novel and could provide many insights to the community. The proposed theoretical formulation also has the potential to be further developed. I enjoy reading this paper and suggest an acceptance.

**Strengths:**

- Good presentation, easy to follow.
- The discussions about the learning dynamics of concept memorization and compositional generalization are cool.
- The three probing methods that verify the claim that the model might learn the concept before generating the correct image are insightful.
- The under-specification discussion might bring practical suggestions to real systems.

**Weaknesses:**

- In Section 3, it is a little hard to understand the relationships between different variables and functions. A diagram like Figure 1 in [2] might be helpful.
- Although the experiments and analysis of the paper are very insightful and persuasive, it is only verified in a relatively simple setting. There are experiments on CelebA, but the number of attributes and their possible values are still kind of small. One or two experiments considering more complex scenarios will strengthen the paper a lot.
- The paper claims that the model usually has the capability to compose the learned concepts earlier than this behavior shows up. How will this finding influence our understanding of (or provide insights to improve) the practical systems? Does this correlate to early stopping (the switching point from underfitting to overfitting)?

**Questions:**

- The legends in Figure 2-bc (so as other learning dynamics figures) are a bit hard to read.
- In Figures 2 and 3, the learning curves exhibit a “zigzag” pattern, i.e., first pointing to the training example sharing the strongest concept value, and then converging to the memorization one or comp-gen one. A similar phenomenon is also observed in a general classification problem discussed in [1]. Is there any similar reason behind these two findings?
- Both the concept signal and sigmoid function are denoted by $\sigma$, which is confusing.
- What are (0.4, 0.4, 0.6) to (0.3, 0.3, 0.7) in line 216 means, normalized RGB values?
- The color bar of Figure 6 might have the wrong legend (it is the percentage of masked $\alpha$, but the bar ranges from 1 to 3.5).
- The paper concludes that  (in line 300) concept learning is a well-controlled phase transition, but the observed behavior can be arbitrarily delayed. Are there any methods that can reduce this delay?
- [2] studies a very similar compositional generalization problem (but in a representation learning setting). It would be helpful to also discuss that in the related work. The discussion of simplicity bias and Kolmogorov complexity in [2] might also be a potential explanation for the observations presented in this paper.

[1] Ren, Yi, Shangmin Guo, and Danica J. Sutherland. "Better supervisory signals by observing learning paths." ICLR 2022

[2] Ren, Yi, et al. "Improving compositional generalization using iterated learning and simplicial embeddings." NeurIPS 2023

**Limitations:**

The experimental setting is kind of simple. But I think it is sufficient for a phenomena explanation (and theory) paper.

---

> ### Author Rebuttal · Authors · 2024-08-07
>
> Dear reviewer o8es,
>
> Thank you so much for thoroughly understanding, and positively evaluating our work. We are glad to hear that you enjoyed our work, and found all of our three main findings to be “quite novel and could provide many insights to the community”. We thank you also for your very insightful and actionable suggestions to robustify our claims. In response to your thoughtful suggestions, we addressed your concerns by adding:\
> (i) a new schematic diagram inspired by [2] Ren, Yi et al.\
> (ii) further experiments in a more complex scenario with three concept variables yielding Fig. R1 and R3\
> (iii) a better grounding into the learning dynamics of compositional generalization(CG) literature [1,2]\
> (iv) a general improvement of figure qualities.
>
> ---
> * **1. Added New Diagram for the framework:** Inspired by [2] Ren, Yi et al., we have updated our framework section of the paper to be easier to understand. We have added a schematic Figure describing the relation of $\mathcal{S}$, $\mathcal{G}$, $\mathcal{M}$, $\mathcal{X}$ in appendix as well. We haven’t been able to fit this new figure in the attached pdf due to space constraints, but this new figure will be included in the final version of this submission.
>
> * **2. Concept space findings generalize to more complex scenarios:** We have now extended our results to more complex scenarios. In Fig. R1, we show that our *findings on hidden emergence and also our probing methodology generalizes to a compositional setup on CelebA*. In this experiment, the model did not show the CG behavior, while latent linear intervention clearly demonstrated the emergence of the capability. We have also generalized our findings on concept signals in Section 4.1 to a 2x2x2 setup with 3 concept variables. We again find that *concept signal controls the speed of learning in scenarios involving more than 2 concept variables*.
>
> * **3. Practical Implications of Early Capabilities:** One implication of the hidden emergence of capabilities is that standard evaluation pipelines on a fixed dataset might be probing behavior, which we show is very seed dependent. Additionally, this motivates studies on enhancing current models by understanding their internals as it suggests that *models can have capabilities which are not elicited unless intervened appropriately*. With your specific point regarding early stopping, we believe that these models do not explicitly over-fit and nor do frontier diffusion models (see [3]), as they are in the under-parametrized regime and usually trained in an online setup. We also do not have evidence of any capabilities getting lost as we train far beyond fitting the training set.
>
> ---
> Questions:
>
> * **Figure 2-bc legend:** Thank you for this feedback! We have made significant revisions to explain the axes better in the captions and to improve the overall quality of the figure.
>
> * **Zigzag Pattern:** Thank you for this reference [1]! Indeed, our current understanding is similar to [1], where higher difficulty corresponds to lower concept signal. In our case a concept is harder when its concept signal is less important to reduce the diffusion loss. We will make sure to cite [1] and discuss the similarity of the mechanisms underlying the shared trend in learning dynamics.
>
> * **Usage of $\sigma$:** Thank you for this suggestion! We updated the sigmoid function to $g$ to avoid confusion.
>
> * **Clarification of line 216:** Yes! We will clarify this in our updated manuscript.
>
> * **Figure 6 Legend** Thank you for this catch, we will edit this.
>
> * **Controlling the phase transition:** We did find out that many hyperparameters control the transition, such as weight decay, weight initialization, optimizer, etc.. However, we did not yet find a clear identifiable relationship which allows any predictive hyperparameter transfer. Testing the generalization of these observations to more complex text-to-image models is a great future direction. Currently, we are interested in following up with a theoretical model based on the concept signal values.
>
> * **Relation to reference [2]:** Thank you for pointing out this work. We will definitely cite and discuss it in the final version! The simplicity bias explanation is very interesting, but it seems non-trivial to define it in our setup at the current stage. We briefly looked into weight decay to add an explicit “simplicity” enforcing term, however this did not qualitatively change our results.
>
> ---
> **Summary**: Thank you again for carefully going through our manuscript and giving us such a concrete and insightful list for improvements! As clarified above, your suggestions inspired us to not only create new schematic figures, but also to run 8 more experiments resulting in 2 new experimental plots. With the above, we hope that we have fully addressed your concerns and now you could strongly recommend acceptance of this paper.
>
> ---
> [3] Kadkhodaie, Zahra, et al. "Generalization in diffusion models arises from geometry-adaptive harmonic representation." arXiv preprint arXiv:2310.02557 (2023).

---

> > ### Comment · Reviewer_o8es · 2024-08-08
> >
> > Thanks very much for the author's hard work. The new results and discussions indeed strengthen the paper. I would like to increase my evaluation to 8. I am looking forward to seeing the final version of the paper.

---

### Official Review · Reviewer_uzj6 · 2024-07-08

**Soundness:** 3
**Presentation:** 3
**Contribution:** 3
**Rating:** 7
**Confidence:** 4

**Summary:**

This paper introduces a new framework, "Concept Space," designed to study the learning dynamics of capability and behavior in generative models. It uses the concept signal to measure the rate of concept learning. They trained a diffusion model on a simplified synthetic dataset to validate their hypotheses that transitions in model capability occur much faster than the corresponding changes in behavior. This work provides empirical insights into understanding and controlling the development of AI models' capability and behavior.

**Strengths:**

1. This paper is well-structured and clearly articulated.
 2. This paper introduces the Concept Space framework and verifies several interesting combinatorial optimization results on synthetic datasets.
3. The experiment setting is meaningful and interesting.

**Weaknesses:**

1. The rationale for modifying the differences in attribute values to represent different levels of concept signals is unclear. How does this relate to the formula $σ_i=|\partial G(\vec{z})/(\partial z_i )|$
The derivation of equation (1) is also not clear.

 2. Uncertainty in generalization: Despite the introduction of the Concept Space framework, there is still uncertainty about the model's ability to generalize to real-world data. While concept learning is easily validated on synthetic datasets, it is unclear how to effectively validate this approach on large-scale datasets and large models. Can you provide some insights?

**Questions:**

N/A

---

> ### Author Rebuttal · Authors · 2024-08-07
>
> Dear reviewer uzj6,
>
> We thank you for your detailed feedback! We are glad you found our concept space framework and the robust emergence of capability insightful and interesting. We are especially happy to hear that our work provides “insights into understanding and controlling the development of AI models' capability and behavior”, as this was one of the main goals of our study. In response to your comments, we have added:\
> (i) An improved description of our framework in Section 3.\
> (ii) experiments on frontier multimodal embedding (CLIP) and generative (Stable Diffusion v2.1) models yielding Fig. R2.
>
> ---
> * **Concept Signal can be altered by color differences:** The concept signal $\sigma_i$ is the absolute change of the data generated when the corresponding concept variable is altered. Here, we are changing the raw RGB difference between the color concept blue and red. Thus ,a rescaling of this difference by $s$ is equivalent to the change of the data generating process $\mathcal{G}\to\mathcal{G}'$, where $|\frac{\partial\mathcal{G}'}{\partial\texttt{color}}|=s|\frac{\partial\mathcal{G}}{\partial\texttt{color}}|$, and thus results in an enhancement of the color concept signal by a factor of $s$. For a visual explanation, please see Fig.7, where the distances are not schematic, but calculated from the dataset as we have access to the data generating process. We have updated Section 3, where we introduce our framework, so that this relation is more clear. We have also moved Fig.7 to the main text. Thank you for prompting this important clarification!
>
> * **Derivation of Equation 1:** Thank you for your question. Equation (1) is a phenomenological model designed to qualitatively reproduce concept learning dynamics, where our goal was to suggest a novel potential space for future theoretical study. The sigmoid function models sudden capability acquisition, while quadratic terms represent the energy landscape guiding the model towards correct concept values. Although not derived from a specific learning process, it introduces a new perspective of defining "concept variables" and describing their evolution via differential equations. We are currently working on a theoretical follow-up to rigorously derive this equation from a concrete model that further substantiates our hypothesis. We will clarify our motivation and limitations in an updated draft.
>
> * **Your feedbacks have inspired us to validate how many of our observations generalize to real-world data scenarios!:** Thank you so much for your insightful suggestions to further explore whether our observations validated in our synthetic setup generalize to real-world data. While, we've made the "simplicity-interpretability" trade-off to derive a set of hypothesis, you are right that it's very important to then go back to realistic scenarios to further validate those hypotheses.
> 1) Latent linear intervention elicits hidden capabilities also in real world facial attribute dataset (CelebA). [Fig R1] In Fig R1a, we first show how diffusion model struggles to compose unseen combination of "female" and "with hat" concepts with naive prompting. Then, in Fig R1b, we demonstrate how applying latent linear intervention elicits this capability as was seen earlier in synthetic dataset!
> 2) Square/cubic structure of concept graphs can be found in CLIP embeddings. We show that the concept space framework is useful to understand the failures and success of CG. *In some cases*, the concept space can be constructed for real diffusion models since it only requires a feature detector (e.g. CLIP), which is much simpler compared to training a generative model. In Fig R2a, we show that CLIP is ready to serve as a feature detector to construct an orthogonal concept space.
> 3) One specific insight from overprompting and latent linear intervention is that one could use model interventions to evaluate model capabilities. In Fig. R2b, we show an explicit example where overprompting results in elicitation of a CG capability. We *speculate* that this could be scaled, as we show in Fig. R2a that CLIP can already serve as an orthogonal feature detector, and the lack of an expected feature can be used to provide feedback onto a method guiding generation by intervention.
> 4) Our work surfaces that controlling concept learning might be more difficult than expected. Our results indicate that *even* In a synthetic setup where the acquisition of capabilities is robust, the model's apparent ability to CG might appear limited, and can show high variance across run seeds. Our contribution can be understood as attributing the high uncertainties observed in CG in real models to the variance of behavior seen in models with equivalent capabilities.
> ---
> * **Summary:** Thank you again for your review. Your comments have motivated us to enhance the introduction of our framework in the manuscript and  to demonstrate scalability to large models and practical setups. As a result, we have managed to show that our intervention strategies could be valid in Stable Diffusion v2.1 and that CLIP embeddings already encompass orthogonal concepts graphs, suggesting a promising direction for scalability. We hope your concerns have been addressed with these clarifications and experiments, and we would be very glad if you could recommend our paper more strongly.

---

> > ### Comment · Reviewer_uzj6 · 2024-08-09
> >
> > Thank you for the detailed response. I really appreciate the idea and motivation behind this paper. The rebuttal has significantly improved the quality of the paper, and I now feel more confident in recommending its acceptance. As a result, I have raised both my score and confidence level.

---

### Official Review · Reviewer_G81k · 2024-07-11

**Soundness:** 3
**Presentation:** 2
**Contribution:** 3
**Rating:** 5
**Confidence:** 3

**Summary:**

This manuscript investigates the distinction between capabilities and behaviors in diffusion models. Using a toy task, the authors analyze at what point during training the model begins generating objects with correct specific features (in particular color and shape). They find that increasing the salience of certain attributes (“concept signal”) makes learning of those attributes faster and further causes the model's compositional generalization to collapse onto the nearest training point after intermediate training times. They replicate qualitative features of these learning curves with a mathematical toy model. They then use different interventions that enable the model to generalize compositionally much earlier than they would otherwise --- the fact that these interventions all yield generalization at the same time suggests a specific time point where the model has learned the underlying capabilities without this necessarily manifesting in behavior. Finally, they use this toy setup to analyze the impact of underspecification on compositional generalization.

**Strengths:**

1. The topic of compositional generalization and concept composition in generative models is important and generally still poorly understood. The authors did a good job motivating this and its connection to the distinction between capability and behavior/competence and performance.
2. The findings in section 4.4 are surprising and notable and the authors replicated this behavior across different model seeds and methodologies (though I have a couple of questions, see weaknesses, point 3).
3. The toy model for underspecification provided helpful intuition and appears to be a particularly simple task that gives rise to such underspecification.
4. The manuscript is generally well-written and the figures are generally easy to follow.
5. Concept memorization is a useful phenomenological finding.

**Weaknesses:**

As noted above, I think there are several interesting findings in this manuscript. However, in its current stage, I believe that it falls short of its stated goals. More specifically:

1. As far as I can tell, sections 4.1-4.3 and section 4.4 get at different forms of "capability." Specifically, the concept space studied in sections 4.1-4.3 distinguishes whether the model is able to generate certain properties of the image (e.g. color and size). In contrast, section 4.4 demonstrates that certain interventions can improve the models accuracy on the compositional task substantially. Since those interventions don't fundamentally change the model's capabilities, this suggests that the model has already learned to do the right thing and these interventions simply surface that capability. It's unclear to me whether that is necessarily apparent from the concept space behavior. Put differently, the model, in principle, could still be performing extremely badly according to the concept space but have already exhibited this transition. (You seem to be getting at a related point in section 6, "Why concept space?" and the supplementary figure, but it's unclear to me what the time of concept of acquisition is. To the extent that it relates the findings in section 4.4 to the findings in the previous sections, I think you'd have to show this across different models rather than using a single example of a model.)
2. I did not understand the role played by equation (1). First, the defined energy landscape always has its minimum (for high t) at $c_1,c_2=1$, so why can these curves tend to different corners of the concept space? Second, I did not understand how this differential equation is grounded in model behavior, as it does not appear to be a simplified learning model. Rather, the goal seems to be to recapitulate the (rough) trajectories in concept space, so I'm not sure what insights are gained from that. In particular, the fact that the model first tends towards the correlated training point (e.g. the small red circle) during learning is built in by the definition of the sigmoidal function.
3. I think the related work section on compositional generalization should provide a better overview of existing insights into the questions you're asking. Right now, you're only citing a number of papers investigating these questions, but I think it would be important to actually give an overview of what these papers are presenting and investigating and how it relates to your own findings. In particular, it's unclear to me what was previously known about the impact of underspecification on compositional generalization.
4. I think it would be important to report standard errors or some sense of deviation across different model seeds. It is still important in my opinion to understand the reliability of these qualitative findings --- e.g. how similar are the concept space curves you're presenting across different initializations?

All in all, I think the paper presents several interesting findings, but, in its current state, leaves unclear how these findings fit together. On the one hand, the concept memorization finding is interesting and works well together with the underspecification finding. I think for both of those findings it would be important to more thoroughly evaluate how reliably the model actually ends up generalizing compositionally (e.g. across different seeds of initialization). In addition, it would also be helpful to give additional mathematical insights (or give an intuition in a different way) into why the observed behaviors emerge, as I don't see the current mathematical model as helpful on this end. On the other hand, the finding in section 4.4 is also interesting, but it remains unclear how it is related to the concept space framework and, if it can be explained in terms of learning both of these capabilities, why the presented intervention mechanism can help the model generalize compositionally.

**Questions:**

1. Could you clarify how sections 4.1-4.3 and section 4.4 are related (see weaknesses, point 1)?
2. Could you clarify how you determined equations (1) and (2) and what we can learn from them (see weaknesses, point 2)?
3. As noted, I think the findings in section 4.4 are really intriguing, so I want to make sure I understand exactly what is going on there. a) Since you're using a binary classifier to assess performance, are the overprompted colors/sizes really identical to the original colors/sizes or is it possible that e.g. the color produced from overprompting is different from the ground-truth color and just more clearly on the correct red/blue side of the hyperplane? It would be useful to see a few examples of generated images here. b) It seems that you're only using one model seed in Figs. 4(c) and (d). For 4(d) you explained that you only used the one with "full capability" (I assume that's the one with an accuracy close to 1.0?). Why did you only use one model for Fig. 4(c) (or are these multiple lines that are just strongly overlapping?)? c) I think the fact that there's only one model that reaches full compositional generalization accuracy qualifies these findings a bit, in particular as it means that the latter methods only present a single sample. Would the other models improve in their performance as well if they were trained for longer? d) I would suggest providing an additional figure where you plot the different curves for each model seed on top of each other, as it is currently a bit difficult to judge how precisely the transition times overlap.
4. I'm not very familiar with the literature on underspecification --- have other papers previously noted its effect on compositional generalization (i.e. the "strawberry"/"yellow strawberry" effect)? If not, I think it'd be worth emphasizing that a bit more --- if yes, it would good to emphasize that as well.

**A couple of minor notes**

L. 29: “pre-training on such models” -> “pre-training of such models”

L. 32: What is the “model experimental systems approach”?

L. 69: Space before citations missing.

L. 96: Is $S$ just the support of the probability distribution?

L. 99: Is $F$ a stochastic function?

L. 117-118: Isn’t $G^{-1}(Y)$ in the concept space, i.e. it should match $z$, not $h$?

Figure 2: Why do the two trajectories for 01 and 10 illustrate concept memorization? I would have thought that this was illustrated by the trajectories for 11 that end up near 01.

L. 247: What does it mean to mask the token? Set it to zero?

**Limitations:**

As the authors acknowledge, they focus on toy synthetic data here. Furthermore, their analysis is largely empirical in nature, leaving unclear the exact reasons why the models generalize or don't generalize. I think the authors have adequately communicated the limitations of their work overall.

---

> ### Author Rebuttal · Authors · 2024-08-07
>
> Dear reviewer G81k,
>
> We thank the reviewer for their very detailed feedback. We are happy you found our findings in 4.4 “surprising and notable”, just the way we felt, and 5. and 4.2 “helpful” and “useful”. In response to your comments and concerns, we have added:\
> (i) A clear definition and clarification of ``Capability” in Sec. 3.\
> (ii) A demonstration that turns in concept space corresponds to emergence of capabilities, yielding Fig. R5 b,c\
> (iii) A substantially more thorough related work section.\
> (iv) A better quantification of uncertainties, yielding Fig. R4 and R5
>
> ---
> * **1. Clarification of Capability in 4.1-4.3 vs 4.4:** The model’s capability is only discussed in 4.4. We do not mention capability in 4.1\~4.3 as we are simply probing its behavior/execution of the task. Nevertheless, we now see that this can be confusing so we clearly defined capability and made this distinction in Sec. 3.\
> Your comment also suggested that the relationship between concept space and capability should be strengthened. We have now added Fig. R5 b,c, which respectively demonstrates that 1) sudden turns in concept space does correspond to emergence of capability to compositionally generalize (CG) and 2) this correspondence is robust across different concept signal levels. We hope this clarifies the relation of section 4.1\~4.3(concept space) and 4.4(hidden emergence).
>
> * **2. Motivation of the phenomenological model was to postulate concept learning dynamics hypothesis:** Eq. (1) is a phenomenological model designed to qualitatively reproduce learning dynamics, requiring different values of c1 and c2 to specify the model generalization point. The initial tendency towards the correlated training point is a result of the sigmoidal function definition, illustrating how different concepts evolve over time. Although the model is not derived from a specific learning process, it introduces the idea of defining "concept variables" and describing their evolution with differential equations. We are currently working on a theoretical follow-up to rigorously derive this equation and further substantiate our hypothesis.
>
> * **3. A more thorough related work section:** Thank you for the feedback! We have significantly expanded the related work section now, including discussion of prior work on CG, concept learning, use of interpretability tools to understand learning dynamics, and distinction between capabilities and behaviors. Related to CG, we briefly note that prior work has primarily focused on impossibility results, i.e., showing whether neural networks can express compositional solutions (e.g. [1]). To our knowledge, there is only one work focusing on learning dynamics of CG in a generative model [2], and does not include any intervention experiments or underspecified setups. Underspecification's influence on CG has been partially explored by a few papers on disentangled representation learning [3], but again these papers focus on possibility results and not learning dynamics---the target of our study. A position paper by Hutchinson et al. [4] mentions challenges in CG due to underspecification, but has a different goal compared to our work.
>
> * **4. Errors quantified and visualized:** Please see Fig. R4,5 for different quantification of errors. In R4, we show that the std. of behavior across seeds is high while it remains very low for probes of capability. In R5a, we visualized the standard error of mean on trajectories in Fig. 2c.
>
> * **5. Clarifications:** The intervention methods are more meant to investigate the model's internal capabilities and show that they emerge consistently before behavior. Although they do improve compositional generalization, that is more of a by product rather than a practical method we introduce.
>
> ---
> Questions:
> * **1, 2, 4:** We hope the replies above address these questions.
> * **3:** a) We have now added example images on top of Figure 4. The generated colors depends on the exact intervention we apply, as we would expect for a steerable model.\
> b,c) We have now repeated the Linear Latent Intervention and Embedder Patching for all 5 seeds. We found LLI to work perfectly on all seeds and EP on 2 seeds. We don't think these affect our findings as one intervention method is *sufficient* to demonstrate model capabilities. Embedder Patching is a more subtle technique (higher bar) since it requires the model to have minimal Embedder-CNN compensating weight changes late in training.\
> d) This is indeed how we have plotted Figs. 4 a, b! If the question was directed towards Figs. 4 c, d, we note we have now added all individual seeds to the plots, as mentioned above.
>
> ---
> Minor Notes
>
> * **L.29,69,117-118:** Thank you for these catches.
> * **L.32:** By “model experimental systems approach”, we mean a small synthetic setup in which we can control the data distribution and probe the resulting model's behavior, allowing one to study questions that are not feasible to do so in real data.
> * **L.96,99:** Yes
> * **Figure 2:** The two gray trajectories are taken from the training data curves to help visualize concept memorization.
> * **L. 247:** What does it mean to mask the token? Set it to zero? Yes, the corresponding element is set to zero.
>
> ---
> * **Summary:** Enormous thanks to the reviewer for pointing out very important points to make our submission more persuasive. We believe our work became more clear and robust thanks to the suggestions. Your comments motivated us to substantially strengthen our theory and related work section. It has also motivated us to quantify uncertainties on our experiments yielding Fig. R4, R5, which are essential to join concept space and hidden emergence, the two big pillars of our work. We believe our manuscript greatly improved in quality, and would be very happy if you could recommend our work more strongly.
>
> ---
> [1] https://arxiv.org/abs/2310.05327
>
> [2] https://arxiv.org/abs/2310.09336
>
> [3] https://arxiv.org/abs/2006.07886
>
> [4] https://arxiv.org/abs/2210.05815

---

> > ### Comment · Reviewer_G81k · 2024-08-08
> >
> > Many thanks to the authors for their careful rebuttal, which has addressed many of my concerns. In particular, I appreciate the authors adding a quantification of uncertainty to their figures and I will increase my score to 5.

---

### Official Review · Reviewer_cQgr · 2024-07-12

**Soundness:** 3
**Presentation:** 1
**Contribution:** 2
**Rating:** 5
**Confidence:** 4

**Summary:**

The paper introduces "concept space", a framework for analyzing the learning dynamics of generative models, focused especially on compositional generalization. The key contributions are:

- Introducing the concept of "concept signal" that governs the rate of concept learning (and in particular determine learning speed), and is the key driver of the generalization dynamics
- Analyzing learning dynamics using the concept space framework on a synthetic dataset of 2D objects.
- Claims about how concept signals shape the geometry of learning trajectories.
- a proposal on interventional protocols to uncover hidden model capabilities.

The paper uses a conditional diffusion model trained on synthetic data to study how concepts like shape, color, and size are learned and composed together. They show that concept signal levels determine learning speed and generalization dynamics, and that model capabilities often emerge before observable behaviors. The paper also explores how underspecification hinders compositional generalization.

**Strengths:**

Originality:

- Concept space and concept signals are a novel lens for analyzing the learning dynamics for diffusion models.
- The claim that "capability consistently emerges before behavior" is interesting and well-backed, and is thus a novel insight into how these models behave over the course of the training.

Quality:

- Operating on the activation space and embedder patching seem like appropriate tests for the claim of testing .
- It shows the intuitive result that model's generations for unseen combinations initially gravitate towards the training class with the strongest concept value.

Clarity:

- The claims are clearly presented with an explanation supported with figures.

Significance:

- The results suggest that the more distinguishable or salient a concept is in the training data (i.e., the stronger its concept signal), the faster the model will learn to recognize and generate that concept. If true for a broader class of generative models, the result could be widely applicable in data design choice for models.

**Weaknesses:**

Significance:

- Overall the setup is too simplistic to have the claims be transferable to a broader set of generative models or even diffusion models. All the experiments use a simplified synthetic dataset with 2D objects and binary concept variables; while good for control, it limits the complexity of the kind of concepts real-world models learn by a massive amount. Real world models are learning more complex, hierarchical, and interdependent concepts, not simply characterizable by things like size or color, and use continuous concept variables rather than binary.
- The concept signal relies on knowing G, the data generating process. In real-world models, the true data generating process is often unknown, concepts might not have clear, differentiable manifestations in the input space.
- The toy model for learning dynamics is based on a simple energy function with sigmoid activation; it's not clear why "concept signals" should be able to capture any non-linear interactions between concepts.

Presentation of the claims:

- The authors use the word "capability" when they really mean the model's internal ability to understand and compose concepts. The wording is quite unclear and should be clarified.
- The graphs are pixelated, low-quality, poorly labeled and hard to understand. In particular, you should explain what the colors and the axis clearly mean in Figure 3, have multiple random seeds analyzed with error bars for overprompting and linear latent intervention in Figure 4, and make Figure 5 (b) higher quality.
- Linear latent intervention assumes linear separability of concepts in the latent space, which breaks down in more complex settings than the synthetic data setting.
- There should've been a higher focus on results on CelebA in the main sections of the paper.

**Questions:**

- Would we expect overprompting to still work when concept manifestations are complex?
- Doesn't the embedder patching technique assume that the final checkpoint has disentangled concepts? Why should we expect to be the case?
- The paper briefly mentions experiments with the CelebA dataset in the appendix. Could the authors elaborate on how well the findings from synthetic data translate to this more complex dataset?

**Limitations:**

- The paper should've compared the concept space framework to other existing methods for representations in learning dynamics (e.g., basic linear probing techniques, representational similarity analysis) as baselines for our understanding to see how much "concept space" improves it.
- The models are fairly simple and small, and the study should further look into whether any of the techniques (e.g. overprompting, linear latent intervention etc.) still continue to hold in setting that more closely resemble real-world conditional diffusion models.

---

> ### Author Rebuttal · Authors · 2024-08-07
>
> Dear reviewer cQgr,
>
> We thank the reviewer for their detailed feedback. We are excited that you found our work provides a novel lens for analyzing learning dynamics of diffusion models, yielding claims that are interesting, clear, and well-backed. In response, we have added:\
> (i) A major experiment on CelebA generalizing our findings to more realistic data, Fig. R1\
> (ii) A test on Stable Diffusion v2.1(SD), Fig R2b\
> (iii) Experiments on a more complex 2x2x2 setup, Fig. R3
>
> ---
> * **1. Simplicity of the setup:** Please note that our goal in this work was to understand how a diffusion model learns various concepts underlying the data-generating process and learns to compose them. The simple setup allowed us to perform a more precise analysis of the model’s learning dynamics and develop novel hypotheses on concept learning. However, our results on CelebA (R1) shows that this simplicity did not bottleneck our claims: our results do transfer to more realistic settings! Orthogonally, prior work on compositional generalization (CG) often uses similar settings (e.g., toy shapes, colors), making our chosen setup a natural starting point.
>
> * **2. Concept Signal relies on G:** Computing precise concept signal values is indeed non-trivial in real data. One approach would be to approximate the derivative of G by training a VAE to compute $dG/dz_i$. However, our primary contribution is a step towards understanding what controls CG when G is known. At the same time, our results show that we can retrospectively establish orders of concept signals (CS) for real data: e.g. we can infer CS for gender is stronger than hat from the CelebA experiment (R1).
>
> * **3. The model for learning dynamics:** We clarify that Fig. 2 shows accuracies over training time, where x and y represents accuracy for color and size. The key insight is that concept signals determine the learning dynamics' trajectory. Fig. 2(c) shows that curves bend toward specific directions reflecting the relative strengths of the concept signals for size and color. To model this behavior, we have defined conditions in Eq. 1: $\sigma_1 > \sigma_2$ and $\sigma_2 < \sigma_1$​. We will provide further clarification on these points in the revised version.
>
> * **4. The definition of capability:** Thank you for the suggestion! You correctly inferred our intended meaning: by capability, we mean “the model's internal ability to understand and compose concepts”. We have now formalized this in Sec. 3 of the paper.
>
> * **5. Figures:** We apologize for the pixelated plots. We updated all figures with higher quality versions. We also labeled all axes and added generated images to aid understanding, showing how the model improves during training (e.g., see R4). As per your suggestions, we have clearly clarified the axes in the caption and labeled the colorbar in Fig. 3. We updated Fig. 4 so that the curves for each seed are clearly visible. We updated Fig. 5b and labeled the color axis as $a$. We thank the reviewer again, as we really value high quality plots especially for a paper focused on sending a qualitative message.
>
> * **6. Linear Latent Intervention(LLI) Breaking Down**\
> The focus of our interventions is mainly to gain understanding of how capabilities are acquired in a model rather than to present a method to improve CG in practical settings. This is also why we presented 3 methods, so that we do not overfit to pitfalls of one specific method. Thus, our methods are *sufficient* to show a model has CG capabilities, but not *necessary* or efficient. Nevertheless, for LLI, we show in R1b that it can generalize to more realistic CelebA data.
>
> * **7. Discussion of Celeb A in main text**\
> Thank you for the suggestion! We moved some of the CelebA results to the main text.
>
> ---
> Questions:
> * **On overprompting(OP):** The short answer is “sometimes" and just as LLI above, this intervention method was *sufficient* to show capability. While there will be cases where OP will not elicit CG, in R2b, we show that OP elicits CG even in SD, demonstrating its generalizability. In fact, SD’s prompt encoder takes brackets [] around words which need to be enhanced, and internally scales the vectors corresponding to these tokens, implementing OP.
>
> * **Assumptions on Embedder Patching:** Yes, and also an even stronger one: weights in MLP and CNNs should not drift after convergence of training. Restating the point made in 6 above, our intervention methods are sufficient but not necessary to show a model’s capability.
>
> * **CelebA Results:** Certainly! We reproduce both concept memorization(R1a) and hidden emergence(R1b) in CelebA. The former checks out the inverse concept signal learning point made above and the latter shows that this synthetic setup captures realistic concept learning well.
>
> ---
> Limitations:
> * **Comparisons:** To our knowledge, the listed tools (e.g., probing) are not standard protocols studying learning dynamics in gen. models. The closest work to ours is Okawa et al. 2023, where the authors used probing to model learning dynamics, but could not identify what determines the order of concept learning, that there is concept memorization, and that there is a hidden emergence of capabilities. Nonetheless, we will add an expanded discussion of these tools in the paper.
>
> * **Generalizability:** We have added 2x2x2 (R3) and CelebA (R1) results to demonstrate that our result, at least, extends to systems which are a little bit more complex.
>
> ---
> * **Summary:** We thank the reviewer again for detailed feedback. The reviewer’s comments motivated us to investigate further into the more realistic CelebA setup, and we are happy to show that our major finding on hidden emergence is well reproduced. We hope our changes and new experiments address the reviewer’s concerns adequately, and would be glad if our paper can be recommended more strongly.
>
> ---
> [1] arxiv.org/abs/2311.03658

---

### Author Rebuttal · Authors · 2024-08-07

Dear Reviewers,

We would like to thank the reviewers for their thoughtful feedback and for recognizing the value of our work. We are pleased that all reviewers found our main contributions, particularly the introduction of the “concept space” framework and the studies of hidden capabilities to be "a novel lens for analyzing the learning dynamics for diffusion models" [R cQgr], "surprising and notable" [R G81k], “meaningful and interesting” [R uzj6] and "quite novel and could provide many insights to the community" [R o8es]. We are trying to understand diffusion models using controlled synthetic experiments and many reviewers pointed out that it is crucial to show how much of the findings generalize to real data/models. This feedback was very helpful and we believe our submission became much more persuasive in this direction with the additional experiments we conducted.

We ran:
* 1 major experiment on CelebA in Fig R1
* 2 experiments with frontier multimodal models in Fig R2
* 27 new experiments with the synthetic setup distributed in Fig. R3, 5
* 8 new probing experiments for Fig. R4
Below are the description of the new experiments and the major changes made to our work. Please find the figures in the attached PDF.

Abbreviations: Compositional Generalization (CG)

---
**New experiments**:
* **[Figure R1] Hidden emergence of capabilities reproduced on CelebA:** Many reviewers asked about the generalizability of our results to a more *realistic* scenario. We trained a conditional diffusion model from scratch on CelebA using the compositional concepts With Hat and Female. Fig R1a shows the concept space dynamics evaluated on CelebA. In this case, CG was harder than the experiment in Fig 9 in the paper, and the model was not able to generate (Female, With Hat), and it was *concept memorizing* on (Female, No Hat). However, Fig R1b shows that latent linear intervention was able to elicit this capability and the model can in fact compose the two concepts it has internalized. Fig R1b clearly shows hidden emergence of capability on a realistic dataset with arbitrarily delayed behavior.

* **[Figure R2] CLIP and Stable Diffusion v2.1 suggests potential scalability of our work:** Reviewers also asked if the *assumptions made and the methods used* are only expected to work in simplistic settings. While we accept that there will be more subtleties in generalizing these to real data, we show promising results. In Fig. R2a we show that CLIP already embeds compositional concepts as a cube with roughly orthogonal axes, addressing the concern that concepts might be non-linearly embedded in realistic models unlike our assumption in the synthetic setup. This also suggests that CLIP can be used as a feature critique to construct a real life concept space. In Fig. R2b, we show that overprompting can be used to elicit CG in Stable Diffusion v2.1, demonstrating that our method does generalize to some extent to frontier models.

* **[Figure R3] Concept signal controls CG on 2x2x2 concept graph:** Reviewers also asked if the results would generalize to more *complex* scenarios than two binary concepts. On this end, we ran experiments with 3 concept variables and reproduced the findings in Fig. 2a, showing that concept signal controls CG timings by affecting concept learning speeds. Fig. R3a shows the concept accuracy versus training time and Fig. R3b shows that the speed of concept learning (defined by the average accuracy up to the final checkpoint) clearly depends on concept signal.

* **[Figure R4] Robust Capability Learning across random seeds:** We confirmed that overprompting and latent linear interventions shows consistent acquisition of CG capabilities across model seeds while the behavior(execution) has high variance. We show that the standard deviation of capability learning curves is negligible compared to behavior. We confirm this behavior on multiple seeds as it is the main result of our paper.

* **[Figure R5] Quantifying uncertainties across random seeds and data distributions:** Our findings on hidden emergence are robust across random seeds and data distributions. We ran multiple seeds of the experiments with different data distributions and confirmed that our results are robust. Fig R4a visualizes the standard error of the mean of the concept space learning dynamics of the models and demonstrates the level of uncertainty in the concept space trajectories. Fig R4b shows that the sudden turn in concept space corresponds to the emergence of a capability across different seeds. Fig R4c shows that this finding is also consistent across different data distributions, varying the level of concept signal.

---
**Major Edits:**
* **Figure Revisions:** We edited most of our figures' qualities and labeling. We made sure all figures had clearly labeled axes and non-pixelated graphs. In particular, for Fig 4, the most important figure of our work, we added many seeds (c,d) and example images (a,b,c,d).
* **Related Works:** We substantially increased the related works section to better ground our paper in the intersection of concept learning, interpretability and cognitive science(competence vs behavior).
* **Framework:** We reformatted the framework section(Sec. 3) so that the term capability is well defined. We explicitly distinguish capability and behavior(execution).
* **CelebA:** We added a section in the main text describing CelebA as many reviewers pointed out the importance of showing generalization of the findings to realistic data.
* **Appendix:** We added all our additional experiments to appendix.

---

### Decision · Program_Chairs · 2024-09-25

**Decision:**

Accept (spotlight)

**Comment:**

This work introduces the idea of "concept space" in the context of conditional generative models and considers the learning process as one that moves from "concept memorization" to generalization. In experiments, the authors demonstrate that, in many cases "capability precedes behavior": models are able to compose concepts under forceful intervention even though they do not successfully generate under normal conditions.

Reviewers found the idea of concept space promising as a lens through which to analyze the dynamics of learning. In particular, the evidence presented that some capability is present in models before it is expressed was deemed highly interesting. Reviewers had some concerns about the limited range of experiments in the model, but the addition of new experiments allayed many of these concerns. I recommend acceptance.